# Sequential Subspace Noise Injection Prevents Accuracy Collapse in Certified Unlearning

## Abstract

Certified unlearning based on differential privacy offers strong guarantees but remains largely impractical: the noisy fine-tuning approaches proposed so far achieve these guarantees but severely reduce model accuracy. We propose sequential noise scheduling, which distributes the noise budget across orthogonal subspaces of the parameter space instead of injecting it all at once. This simple modification mitigates the destructive effect of noise while preserving the original certification guarantees. We extend the analysis of noisy fine-tuning to the subspace setting, proving that the same $(\varepsilon, \delta)$ privacy budget is retained. Empirical results on image classification benchmarks show that our approach substantially improves accuracy after unlearning while remaining robust to membership inference attacks. These results show that certified unlearning can achieve both rigorous guarantees and practical utility.

## 1 Introduction

Machine unlearning refers to the task of transforming a machine learning model so that the influence of a specified subset of training data is removed.

The problem has gained attention (Nguyen et al., 2024), partly due to the European Union's General Data Protection Regulation (GDPR) (European Commission, 2016) and the associated "right to be forgotten" (Hoofnagle et al., 2019), which grants individuals the right to request the erasure of their personal data. Beyond legal requirements, unlearning is also practically relevant: it can be used to remove private or sensitive information, or to mitigate the effect of poisoned or maliciously injected data. These applications highlight the need for efficient and reliable unlearning techniques.

The most direct baseline is retraining from scratch on the retained data, but this is typically computationally infeasible. Existing methods are commonly divided into three categories: exact, certified, and empirical (Guo et al., 2020; Neel et al., 2021; Fan et al., 2024; Jia et al., 2023). Exact and certified approaches offer formal guarantees but usually rely on restrictive assumptions or architectural changes, whereas empirical methods lack guarantees and instead rely on evaluation tools such as Membership Inference Attacks (MIA) (Kurmanji et al., 2023; Jia et al., 2023).

Among certified approaches, those inspired by differential privacy are particularly notable (Dwork & Roth, 2014; Balle et al., 2020; Liu et al., 2023; Allouah et al., 2025), yet they often struggle to preserve model utility: noise is injected into the parameters to guarantee forgetting, yet even after fine-tuning the model typically fails to regain its original performance. A recent line of work, in particular the noisy fine-tuning approach based on gradient clipping (Koloskova et al., 2025), constructs $(\varepsilon, \delta)$-certification for *arbitrary models and loss functions*. However, empirical results indicate that even in relatively small-scale settings the required noise causes substantial accuracy degradation, limiting the practical applicability of the method. For instance, on CIFAR-10 with ResNet-18, the noisy fine-tuning shows a significant drop in test accuracy during the unlearning phase and fails to recover afterwards (Figure 1).

To address this limitation, we propose a block-wise variant of noisy fine-tuning that preserves accuracy more effectively while retaining certified guarantees. The key idea is to partition the parameter space into orthogonal subspaces (for example, but not necessary, corresponding to different layers) and to apply the unlearning procedure sequentially across them, distributing the effect of noise over time rather than injecting it all at once.

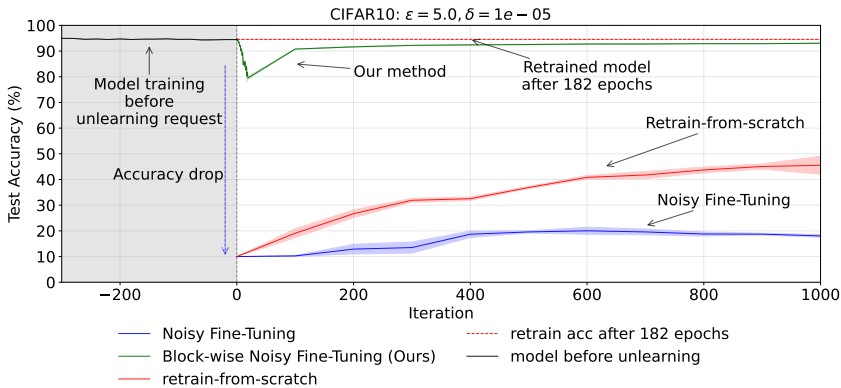

Figure 1: **Severe accuracy drop under noisy fine-tuning.** On CIFAR-10 with ResNet-18, standard NFT test accuracy drops sharply from 98% to below 20% once the unlearning begins, and does not recover even after 1000 subsequent fine-tuning steps.

We further assume proximity between the fully trained and retrained models (under the same coupling of randomness). Under this assumption, our method maintains certified guarantees while achieving substantially better post-unlearning accuracy. Empirically, we show that the resulting models both preserve utility and successfully forget the designated data, as confirmed by standard empirical unlearning evaluations.

**Motivation.** The central challenge in certified unlearning is to design methods that *simultaneously* preserve rigorous differential-privacy–based guarantees and avoid the severe accuracy degradation caused by heavy clipping and noise in current DP-inspired approaches.

Our work makes the following **contributions**:

- **Method.** We introduce *Sequential Subspace Noise Injection* (also referred to as *block-wise noisy fine-tuning*): the model parameters are partitioned into $k$ orthogonal subspaces, and only one block is updated per step. This sequential schedule distributes the required Gaussian noise across blocks and iterations, reducing per-step distortion compared to injecting noise into all parameters at once.
- **Theory.** We extend the certification analysis of noisy fine-tuning to our block-wise schedule and show that it preserves the same $(\varepsilon, \delta)$ budget. Furthermore, under a mild proximity assumption on retrained models, the initial model clipping step can be omitted and the constants in the guarantee improved, without weakening privacy.
- **Practice.** In experiments on MNIST and CIFAR-10 with standard architectures, our method consistently reduces the post-unlearning accuracy drop compared to certified baselines, both for random and class-wise deletions, while maintaining robustness against membership inference attacks (MIA).

In summary, our approach preserves formal certified unlearning guarantees while substantially mitigating the utility loss that has so far limited differentially private methods in practice.

## 2 PRELIMINARIES AND PROBLEM STATEMENT

In this section, we fix notation and recall the standard definitions used in the rest of the paper.

**Setup.** A (possibly randomized) learning algorithm $\mathcal{A}$ maps a dataset $\mathcal{D}$ to model parameters $\hat{\mathbf{x}} \in \mathbb{R}^d$, i.e., $\hat{\mathbf{x}} = \mathcal{A}(\mathcal{D})$. A deletion request specifies a subset $\mathcal{D}_f \subseteq \mathcal{D}$ to be removed; the retained data are $\mathcal{D}_r := \mathcal{D} \setminus \mathcal{D}_f$. An unlearning mechanism $\mathcal{U}$ takes $(\hat{\mathbf{x}}, \mathcal{D}, \mathcal{D}_f)$ and, using randomness, outputs updated parameters $\tilde{\mathbf{x}} = \mathcal{U}(\hat{\mathbf{x}}, \mathcal{D}, \mathcal{D}_f)$.

We adopt the definition from Koloskova et al. (2025), where the notion of certified approximate unlearning is introduced, building on an analogy with differential privacy.

**Definition 1** (($\varepsilon, \delta$)-unlearning (Koloskova et al., 2025)). *Let $\varepsilon \geq 0$, $\delta \in [0, 1]$. We say that $\mathcal{U}$ is an ($\varepsilon, \delta$)-unlearning algorithm for $\mathcal{A}$ if there exists a certifying algorithm $\bar{\mathcal{A}}$ such that, for any forget*

dataset $\mathcal{D}_f \subset \mathcal{D}$ and any observation $O \subset \mathbb{R}^d$,

$$\Pr[\mathcal{U}(\mathcal{A}(\mathcal{D}), \mathcal{D}, \mathcal{D}_f) \in O] \leq e^\varepsilon \Pr[\bar{\mathcal{A}}(\mathcal{D} \setminus \mathcal{D}_f) \in O] + \delta,$$
$$\Pr[\bar{\mathcal{A}}(\mathcal{D} \setminus \mathcal{D}_f) \in O] \leq e^\varepsilon \Pr[\mathcal{U}(\mathcal{A}(\mathcal{D}), \mathcal{D}, \mathcal{D}_f) \in O] + \delta. \tag{1}$$

Note that, by definition, $\bar{\mathcal{A}}$ may be any algorithm. Following Koloskova et al. (2025), we focus on guarantees with respect to $\bar{\mathcal{A}}(\mathcal{D} \setminus \mathcal{D}_f) = \mathcal{U}(\mathcal{A}(\mathcal{D}_r), \mathcal{D}_r, \varnothing)$.

In other words, we study the closeness between the distribution of the unlearning outcome applied to a model trained with the forget set and that of the reference model trained without it. Importantly, this definition provides a framework for reasoning about the indistinguishability of the two models, but it does not by itself guarantee that the resulting model preserves high accuracy.

We build upon the noisy fine-tuning method introduced by Koloskova et al. (2025), which is inspired by the standard DP-SGD algorithm (Abadi et al., 2016) and applied only to the retained data $D_r$. The method combines gradient clipping with Gaussian noise injection and is defined as follows:

**Definition 2** (Noisy fine-tuning (Koloskova et al., 2025))**.**

$$\mathbf{x}_0 = \Pi_{C_0}(\hat{\mathbf{x}}), \tag{2a}$$
$$\mathbf{x}_{t+1} = \mathbf{x}_t - \gamma\big(\Pi_{C_1}(g_t) + \lambda\mathbf{x}_t\big) + \boldsymbol{\xi}_{t+1}. \tag{2b}$$

where $\mathbf{x}_t$ are the parameters at iteration $t$, $g_t$ is the gradient at step $t$ (computed on $\mathcal{D}_r$), $\gamma > 0$ is the learning rate, $\lambda \geq 0$ is the weight decay parameter, $\boldsymbol{\xi}_{t+1} \sim \mathcal{N}(0, \sigma^2 I_d)$ is Gaussian noise, and $\Pi_{C_0}, \Pi_{C_1}$ are clipping operators with radii $C_0, C_1 > 0$, defined as $\Pi_C(\mathbf{v}) := \mathbf{v} \cdot \min\big\{\frac{C}{\|\mathbf{v}\|}, 1\big\}$.

For comparison, we also introduce notation for retrained models. Let $\hat{\mathbf{x}}' = \mathcal{A}(\mathcal{D}_r)$ denote the model parameters obtained by training on the retained dataset $\mathcal{D}_r = \mathcal{D} \setminus \mathcal{D}_f$ with the same algorithm and a suitably *fixed* coupling of randomness (e.g., matched random seeds or noise schedules). Accordingly, let $\mathbf{x}'_t$ denote the iterates produced by applying the updates from to the model $\hat{\mathbf{x}}'$.

## 3 ALGORITHM MOTIVATION

### 3.1 RETRAINED MODEL LOCALIZATION

The goal of this subsection is to explain why the standard analysis of NFT with model clipping is overly conservative and inevitably leads to poor utility. In particular, we show via a simple thought experiment that under this analysis NFT cannot be both faster than full retraining and guarantee good accuracy. This observation motivates our later modification, where the initial model clipping with the clipping radius $C_0$ is replaced by the tighter closeness parameter $\Delta(\rho)$.

**Setup.** Let $T_{\text{retrain}}$ denote the minimal number of training steps required to reach good accuracy when retraining a model from scratch on the retain set $D_r$. By construction, any procedure that achieves comparable accuracy in fewer than $T_{\text{retrain}}$ steps would constitute a faster retraining algorithm, contradicting the definition of $T_{\text{retrain}}$.

In the standard NFT analysis, the number of unlearning steps $T$ is fixed in advance as a function of the privacy budget $(\varepsilon, \delta)$, the learning rate, and the clipping parameters. Crucially, $T$ does *not* depend on the initialization of the model parameters. Moreover, the certified guarantee is formulated in the worst case: it requires NFT to produce indistinguishable outcomes for *any* two initializations within the clipping ball of radius $C_0$.

**Thought experiment.** Suppose that NFT, when started from the fully trained model $\hat{\mathbf{x}}$, achieves accuracy at least $\alpha$ with probability $p$ after $T$ steps, where $T \ll T_{\text{retrain}}$. Because the guarantee is independent of the initialization and holds for any model inside the clipping ball, we may equally well start NFT from a randomly initialized model $\mathbf{x}_{\text{init}}$. If, for our loss function, closeness of the weight distributions implies closeness of model performance, then after the same $T$ steps this procedure must produce a model whose performance is close to that of NFT started from $\hat{\mathbf{x}}$. By assumption, this implies that retraining from scratch can reach accuracy close to $\alpha$ in only $T$ steps.

This directly contradicts the definition of $T_{\text{retrain}}$, which is the minimal number of steps required for full retraining. Hence NFT cannot at the same time (i) guarantee good performance and (ii) be strictly faster than full retraining under the worst-case clipping analysis.

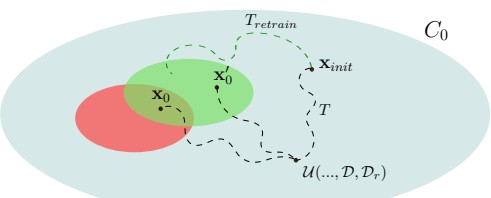

Figure 2: **Illustration of the intuition behind the negative result for Noisy Fine-Tuning.** The illustration shows fully-trained model $\mathbf{x}_0$, retrained model $\mathbf{x}_0'$ and $\mathbf{x}_{\text{init}}$ after model clipping. Unlearning trajectories requires $T < T_{\text{retrain}}$ steps. However, we need at least $T_{retrain}$ steps of unlearning on the $\mathbf{x}_{\text{init}}$ to reach a good quality (the green region). Therefore, results of unlearning ($T$ step from $\mathbf{x}_{\text{init}}$) cannot obtain a good quality.

**Implication.** The contradiction highlights that the clipping-based guarantee is *too strong*: it enforces indistinguishability across all possible initializations, even completely random ones, which is not required by the definition of certified unlearning. In practice, the distance between the fully trained model $\hat{\mathbf{x}}$ and the retrained model $\hat{\mathbf{x}}'$ is much smaller than the clipping radius $C_0$. We therefore replace the initial clipping with the clipping radius $C_0$ by a high-probability bound $\Delta(\rho)$ on this distance, yielding significantly tighter and more practical guarantees in the subsequent analysis.

**Definition of Initial Discrepancy.**

**Definition 3** (High-Probability Initial Discrepancy). *For any failure probability $\rho \in (0,1)$, we define the* initial discrepancy $\Delta(\rho)$ *between the fully trained model $\hat{\mathbf{x}}$ and the retrained model $\hat{\mathbf{x}}'$ as the smallest value satisfying*

$$\Pr\big[\|\hat{\mathbf{x}} - \hat{\mathbf{x}}'\| \leq \Delta(\rho)\big] \geq 1 - \rho. \tag{3}$$

**Remark 1.** *Discrepancy $\Delta(\rho)$ for $\rho = 1$ is connected to a sensitivity of the model's output. However, we only considering particular joint distribution $(X, X')$ while sensitivity compares overall distance maximum overall possible $X$ and $X'$.*

**Remark 2.** *Different couplings may yield different values of $\Delta(\rho)$; we take the coupling to be part of the definition of the certifying procedure and hold it fixed throughout the analysis.*

**Discussion.** Removing model clipping and replacing $C_0$ with $2\Delta(\rho)$ in parameter calculation produces tighter noise calibrations and sharper guarantees whenever the two initial models are close, but also obliges us to estimate or bound $\Delta(\rho)$ in advance. This shift changes the guarantee from "for *any* pair of initializations in the ball" to "for the particular (distribution of) initializations arising from full-data and retained-data training"—which is exactly the quantity required by the certified unlearning definition.

For the experimental section, we treat $\Delta(\rho)$ as a tunable parameter and do not attempt to estimate $\rho$; the results should be interpreted under this assumption. However, even without access to the exact value of $\rho$ for particular value of $\Delta$, in practice we observe higher MIA efficacy.

## 3.2 Noise distribution

Skipping the initial clipping step as in Section 3.1 relaxes the requirement to hold for every initialization in the $C_0$-ball, but by itself does not prevent accuracy loss. Empirically, we still observe that during the unlearning phase accuracy often drops sharply and cannot be fully recovered by subsequent fine-tuning. The reason is that the injected noise is large enough to dominate the gradient signal across all parameters.

We now formalize this limitation with a lower bound on the per-step noise level. The bound is expressed in terms of $(q, \varepsilon^{\text{rényi}})$-Rényi Differential Privacy (RDP) (Mironov, 2017), where $\varepsilon^{\text{rényi}}$ denotes the privacy loss at order $q > 1$. As standard, an RDP guarantee can be converted into an $(\varepsilon, \delta)$-guarantee (Definition 1) via

$$\varepsilon = \varepsilon^{\text{rényi}} + \frac{\log(1/\delta)}{q-1}.$$

**Theorem 1** (Per-step noise lower bound). *Let $\gamma > 0$ be the learning rate and $\lambda \geq 0$ the weight decay parameter, with $\gamma\lambda < 1$. Consider Noisy Fine-Tuning with gradient clipping radii $C_0, C_1 > 0$*

*and Gaussian perturbations, certified via Rényi DP. Then any noise scale $\sigma$ that enables $(\varepsilon, \delta)$-unlearning must satisfy*

$$\sigma^2 \begin{cases} \geq \gamma(2 - \gamma\lambda)\frac{2q}{\varepsilon^{r\acute{e}nyi}}\left(2 - \frac{\lambda C_0}{C_1}\right)C_0 C_1, & \text{if } \frac{\lambda C_0}{C_1} \in (0, 1), \\[2ex] > \gamma(2 - \gamma\lambda)\frac{2q}{\varepsilon^{r\acute{e}nyi}}\frac{C_1^2}{\lambda}, & \text{if } \frac{\lambda C_0}{C_1} \in [1, \infty). \end{cases} \tag{4}$$

*This inequality holds for* any *number of unlearning steps $T$.*

*The full dependence of the minimal step count $T(\sigma^2)$ on the noise level is derived in Appendix D.1. In particular, in the first regime $\frac{\lambda C_0}{C_1} \in (0, 1)$ and for the minimal feasible noise $\sigma_{\min}^2$, the required number of steps is*

$$T(\sigma_{\min}^2) = \frac{\log\left(1 - \frac{\lambda C_0}{C_1}\right)}{\log(1 - \gamma\lambda)}.$$

**Interpretation.** Theorem 1 is a *necessary condition within the proof framework of Koloskova et al. (2025)*. If $\sigma^2$ falls below the stated threshold, the divergence bound (Theorem A.9 in their work) cannot be satisfied, and the mechanism cannot be certified as $(\varepsilon, \delta)$-unlearning by this analysis. This does not rule out that other algorithms or analyses might achieve valid guarantees with smaller noise: the result is proof-technique limited, not an information-theoretic impossibility.

**Intuition.** Even at the minimal feasible noise level $\sigma_{\min}$, every coordinate receives Gaussian noise at each step, so the noise vector typically has $\ell_2$-norm about $\sigma_{\min}\sqrt{d}$. For networks with millions of parameters, this perturbation causes the additive noise to dominate the *clipped* update across many coordinates, explaining the sharp accuracy degradation observed in practice. The two regimes $\frac{\lambda C_0}{C_1} < 1$ versus $\frac{\lambda C_0}{C_1} \geq 1$ reflect whether gradient clipping or weight decay dominates the dynamics.

**Remark 3** (Closeness assumption)**.** *The statement of Theorem 1 remains valid if the initial model clipping is skipped and the closeness assumption (Section 3.1) is used instead. In that case, all occurrences of $C_0$ can be replaced by $2\Delta(\rho)$.*

Taken together, the theorem highlights why NFT struggles in over-parameterized models: certification forces the injection of noise into *all* coordinates at each step. This motivates our adaptation based on *sequential subspace injection*, where the same noise budget is redistributed across orthogonal subspaces instead of applied globally.

## 4 ALGORITHM

### 4.1 BLOCK-WISE NOISY FINE-TUNING

In our approach, we fix the number of subspaces (blocks) $k \in \mathbb{N}$ and partition the weight space $\mathbb{R}^d$ into $k$ mutually orthogonal components.

To construct this partition, choose integers $r_1, \ldots, r_k \geq 0$ with $\sum_{i=1}^k r_i = d$ and build matrices $A_i \in \mathbb{R}^{d \times r_i}$ whose columns are orthonormal. Define

$$A := [\, A_1 \; \cdots \; A_k \,] \in \mathbb{R}^{d \times d}, \qquad \text{so that } A^\top A = I_d. \tag{5}$$

Then the subspaces are $V_i := \text{span}(A_i)$; they are mutually orthogonal and $\mathbb{R}^d = \bigoplus_{i=1}^k V_i$.

**Proposition 1.** *Each weight vector $W \in \mathbb{R}^d$ can then be uniquely decomposed as*

$$W = \sum_{i=1,\ldots,k} A_i B_i, \tag{6}$$

and we provide a formal proof of this statement in the Appendix D.

The overall procedure is summarized in Algorithm 1. At each stage, we project the parameters onto one of the orthogonal subspaces, apply noisy fine-tuning without model clipping restricted to this subspace, and finally run several standard fine-tuning steps on the full model.

---

**Algorithm 1** Block-wise noisy fine-tuning for unlearning

---

**Require:** model $\hat{\mathbf{x}}$, parameters $\gamma, \lambda, \Delta(\rho), C_1$, number of blocks $k$ and privacy budget $(\varepsilon, delta)$.
1: Define projection matrices $A_1, \ldots, A_k$.
2: Decompose the model weights $\hat{\mathbf{x}}$ as $\hat{\mathbf{x}} = \sum_{i=1}^{k} A_i B_i$ (the $A_i$ are fixed and not trained).
3: **for** $i = 1, \ldots, k$ **do**
4:     Freeze all parameters except $B_i$.
5:     Calculate noise variance $\sigma^2$ and number of steps $T$ using formula from Theorem 1 with $C_0$ replaced by $2\Delta(\rho)$ (Remark 3).
6:     Apply noisy fine-tuning *without model clipping* with respect to $B_i$.
7: **end for**
8: Run several standard fine-tuning steps with all model parameters unfrozen.

---

**Remark 4.** *At each sequential step, noise is injected only into the parameters of a single subspace of dimension $r_i$, rather than into the full parameter vector of dimension $d$. When blocks have equal size ($r_i = d/k$), this corresponds to perturbing only a $\frac{1}{k}$-fraction of the coordinates per step. Thus, although the noise variance $\sigma^2$ is unchanged, the* total *perturbation per step is smaller, which helps preserve model utility.*

### 4.2 THEORETICAL GUARANTEES

We also prove that our algorithm preserves theoretical unlearning guarantees.

**Theorem 2.** *For the decomposition $W = \sum_{i=1}^{k} A_i B_i$ and noisy fine-tuning algorithm with parameters $(\varepsilon_i, \delta)$ there is $(\varepsilon, \delta)$ unlearning guarantee, where*

$$\varepsilon = \sum_{i=1}^{k} \varepsilon_i^{rényi} + \frac{\log(1/\delta)}{q-1} = \sum_{i=1}^{k} \varepsilon_i - (k-1)\frac{\log(1/\delta)}{q-1}. \tag{7}$$

We extend the proof of Koloskova et al. (2025), which relies on a sequence of privacy amplification inequalities with shifted Rényi divergence (Balle et al. (2020)) as the key tool. Our adaptation handles the multi-dimensional shift scenario induced by the block decomposition. Specifically, we generalize the definitions of Wasserstein distance, shifted Rényi divergence, and the Shift Reduction Lemma (Feldman et al., 2018), which bounds the divergence under Gaussian noise. We provide our adaptation and proof of the Shift Reduction Lemma, with the full proof given in Appendix D.2

**Definition 4** (Decomposition gap). *Let $A_i$ for $i = 1, \ldots, k$ be fixed set of matrices as defined in 5. Let*

$$W = \sum_{i=0}^{k} A_i B_i, \qquad W' = \sum_{i=0}^{k} A_i B_i'.$$

*We define the* decomposition gap *between $W$ and $W'$ as*

$$\mathcal{G}(W, W') := (z_0, \ldots, z_k), \qquad z_i := \|B_i - B_i'\|.$$

*For two such vectors $z^{(1)}$ and $z^{(2)}$, we write $z^{(1)} \preceq z^{(2)}$ if the inequality holds coordinate-wise, i.e., $z_i^{(1)} \leq z_i^{(2)}$ for all $i$.*

This leads to the definitions of the $\infty$-Wasserstein distance and the shifted Rényi divergence.

**Definition 5** (Decomposed Wasserstein distance). *We say that $W_d(\mu, \mu') \preceq (z_1, \ldots, z_k)$ if there exists a coupling $w \in \Gamma(\mu, \mu')$ such that, almost surely for $w \sim (x, y)$,*

$$\mathcal{G}(x, x') \preceq z$$

**Definition 6** (Decomposed shifted Rényi divergence). *For any $z \in \mathbb{R}_+^k$, $q \geq 1$, and two distributions $\mu, \nu$ defined on $\mathbb{R}^d$, we define*

$$D_q^{(z)}(\mu \,\|\, \nu) := \inf_{\mu': W_d(\mu', \mu) \preceq z} D_q(\mu' \,\|\, \nu). \tag{8}$$

The new divergence retains many properties of the original. In particular, with zero shift it reduces to the standard Rényi divergence. Moreover, the Shift Reduction Lemma can be adapted to bound the divergence before and after adding Gaussian noise.

**Lemma 1** (Decomposed Shift Reduction Lemma for Gaussians). *Let $q \geq 1$, $z, a \geq 0$, and $X, Y$ be arbitrary random variables. For any matrix $A_i$ from 4 If $\xi, \xi' \sim \mathcal{N}(0, \sigma^2 I_{r_i})$ with $\sigma > 0$, then*

$$D_q^{(z)}(X + A_i\xi \,\|\, Y + A_i\xi') \;\; \leq \;\; D_q^{(z+ae_i)}(X \,\|\, Y) + \frac{qa^2}{2\sigma^2}. \qquad (9)$$

*Proof.* The original proof for unshifted case is adapted by modifying the first inequality step. In particular, instead of considering $(X + W, -W + \xi)$ and $(Y, \xi)$, we consider $(X + A_iW, -W + \xi)$ and $(Y, \xi)$, so as to ensure the shift $(0, \ldots, a, \ldots, 0) = ae_i$. Indeed, $X + A_i\xi$ and $Y + A_i\xi$ can be obtained from $(X + A_iW, -W + \xi)$ and $(Y, \xi)$ by post-processing $f(x, y) = x + A_iy$.

For the non-zero shift $z$, we adapt the proof by redefining $W_1$. Rather than the original choice $W_1 = h_z(W)$ (with $h_z(x) = x$ if $|x| \leq z$ and $h_z(x) = \frac{x}{|x|}z$ otherwise), we set

$$W_1 = \sum A_i h_{z_i}(B_i).$$

In this case, we observe that $W_1$ satisfies $G(W_1, 0) \preceq z$. Moreover, $G(W, W_1) \preceq ae_i$ whenever $G(W, 0) \preceq z + ae_i$. The remainder of the proof then follows directly from the original argument. $\quad\square$

We next show that, under this construction, using the same noise level $\sigma^2$ is equivalent to distributing the noise across multiple blocks.

**Proposition 2.** *Let $W = \sum_{i=1}^{k} A_iB_i$ be a decomposition from 6. Adding $\zeta \sim \mathcal{N}(0, \sigma^2 I_d)$ directly to $W$ is equivalent to adding independent $\zeta_i \sim \mathcal{N}(0, \sigma^2 I_{r_i})$ to each block $B_i$, in the sense that the resulting noisy weight distributions coincide.*

*Proof.* Each term $A_i\zeta_i$ is Gaussian with covariance $\sigma^2 A_i A_i^\top$. Since the noises are independent and the blocks $A_i$ span $\mathbb{R}^m$ orthogonally, the sum is Gaussian with covariance $\Sigma = \sum_i \sigma_i^2 A_i A_i^\top = \sigma^2 I_m$, which reproduces isotropic i.i.d. noise on $W$. $\quad\square$

### 4.3 Subspace design strategies

In this section, we discuss several concrete strategies for constructing the matrices $A_1, \ldots, A_k$.

**Random orthonormal matrix.** We generate a random orthonormal basis $[A_1, \ldots, A_k] \in \mathbb{R}^{m \times m}$ by sampling a Gaussian matrix and orthogonalizing it. Intuitively, this distributes both noise and potential degradation evenly across blocks: if unlearning harms one block, the others can compensate and help preserve accuracy.

Since weight dimensionality can be very large, in practice we construct $A_1, \ldots, A_k$ separately for each layer. For a layer with weight matrix of size $m \times n$, we apply the same procedure locally, which requires an additional $O(m^2)$ memory per layer.

While Theorem 2 shows that Rényi $\varepsilon^{\text{rényi}}$ accumulates additively, we demonstrate that splitting into equal-dimensional random subspaces maintains the overall budget, even though the number of unlearning steps increases by a factor of $k$.

**Corollary 1** (from Theorem 2). *If the weights are split into $k$ approximately equal blocks, then the method guarantees an $(\varepsilon, \delta)$-budget with a total of $k \cdot T$ steps, where $T$ is the number of steps for the baseline algorithm without decomposition. These steps are computationally lighter, and in practice the total can be smaller than $k \cdot T$.*

As in the original noisy fine-tuning method, we add several standard fine-tuning steps after unlearning. Thus, for small $T$ the overall runtime is close to that of the baseline, while the block-wise method achieves better and more stable accuracy.

**Random permutation matrix.** To further reduce memory, one may use a permutation matrix $A$, again applied layer-wise. In this case each coordinate of the weight vector belongs to exactly one block. For a layer of size $m \times n$, the memory cost is only $O(m)$.

**Layer-wise grouping.** A random matrix is not always the most suitable choice. In many architectures, the head and body evolve differently during training, with the head typically more flexible

than the body. This motivates assigning different unlearning parameters to these groups of weights, or designing blocks that correspond to entire subsets of layers. This approach requires no additional memory.

# 5 EXPERIMENTS

**Hyperparameters.** We set $\Delta(\rho) = 0.01$ for *random 10% deletion* and $\Delta(\rho) = 0.05$ for *classwise deletion*. In our comparison and auditing experiments (MIA and accuracy on $\mathcal{D}_f$), these values were sufficient to remove identifiable signal. For all remaining hyperparameters we perform a small grid search over a predefined subset and select the best configuration on a held-out split of $\mathcal{D}_r$. Further implementation details are in Appendix E.1, and our code is available in Appendix B.

In this section we empirically evaluate Block-wise Noisy Fine-Tuning (Block-wise NFT). We first compare our method against two common baselines: *retraining from scratch* and a variant of *Noisy Fine-Tuning* in which *the initial clipping is replaced by the discrepancy* $\Delta(\rho)$ (NFT). Then compare to several empirical unlearning methods using Membership Inference Attack (MIA) auditing.

**Benchmarks, models, and scenarios.** We evaluate on MNIST (LeCun et al., 1998) with a fully connected network of 4.36M parameters (architecture in Appendix E.1) and on CIFAR-10 (Krizhevsky, 2009) with a standard ResNet-18 (He et al., 2016). We consider two deletion settings: *random 10% deletion* and *classwise deletion*, with additional results in Appendix E.

**Procedure.** For NFT-based methods we fix the privacy (unlearning) budget $(\varepsilon, \delta)$ per plot, with $\delta = 10^{-5}$ held constant, and show separate plots for different $\varepsilon$ values. For each setting, we compute the minimal feasible noise scale $\sigma$ (via Theorem 1), derive the corresponding number of steps $T$, and run unlearning for these $T$ steps (Block-wise NFT applies the procedure sequentially across blocks). After unlearning, we apply standard fine-tuning.

We limit the *total* iteration number for the entire unlearning process at 1000 iterations, which corresponds to at most 1.5 epochs on both MNIST and CIFAR-10 at batch size 64 with 90% of the data. In practice, methods typically reach their peak accuracy well before the end of fine-tuning.

Unless stated otherwise, we use the *Random Blocks* construction for $A$ (as described in 4.3); alternative block structures are evaluated in the appendix. All results are averaged over 5 independent runs to ensure statistical reliability.

## 5.1 BLOCK-WISE NFT VS. NFT

Figure 3 shows results for the random 10% deletion task. Block-wise NFT is consistently more stable: the accuracy drop during the unlearning phase is smaller, and recovery under fine-tuning is stronger. For instance, at the tight budget $\varepsilon = 0.5$ on MNIST, fails to recover even after fine-tuning, while Block-wise NFT retains non-trivial accuracy.

The trajectories are smoother, suggesting that distributing noise across subspaces better preserves retained data. Increasing the number of blocks ($2 \rightarrow 4 \rightarrow 10$) further reduces early accuracy loss, with $k = 10$ giving the best results.

## 5.2 COMPARISON TO EMPIRICAL METHODS

Table 1 reports results for the task of forgetting class 5. We compare Block-Wise NFT to retraining, fine-tuning (FT) (Warnecke et al., 2023b), gradient ascent (GA) (Thudi et al., 2022), influence unlearning (IU) (Koh & Liang, 2017), and sparsity-based approaches (SaLUN (Fan et al., 2024), $\ell_2$-sparsity (Jia et al., 2023)). Baselines were reproduced using the official SaLUN repository.

Evaluation uses standard metrics: unlearning accuracy (UA, defined as $1 - \text{Acc}(\mathcal{D}_f)$), test and retain accuracy (TA and RA), membership inference attack score (MIA) (Jia et al., 2023) (see Appendix G for details), and run-time efficiency (RTE, minutes).

**Observations** Block-Wise NFT achieves UA=100% and MIA=100%, fully matching retraining and surpassing empirical baselines such as FT and GA. The perfect MIA score indicates that our certified approach is robust to MIAs, ensuring that forgotten data leaves no exploitable trace. Moreover, among methods with full forgetting, Block-Wise NFT requires the lowest training effort (RTE).

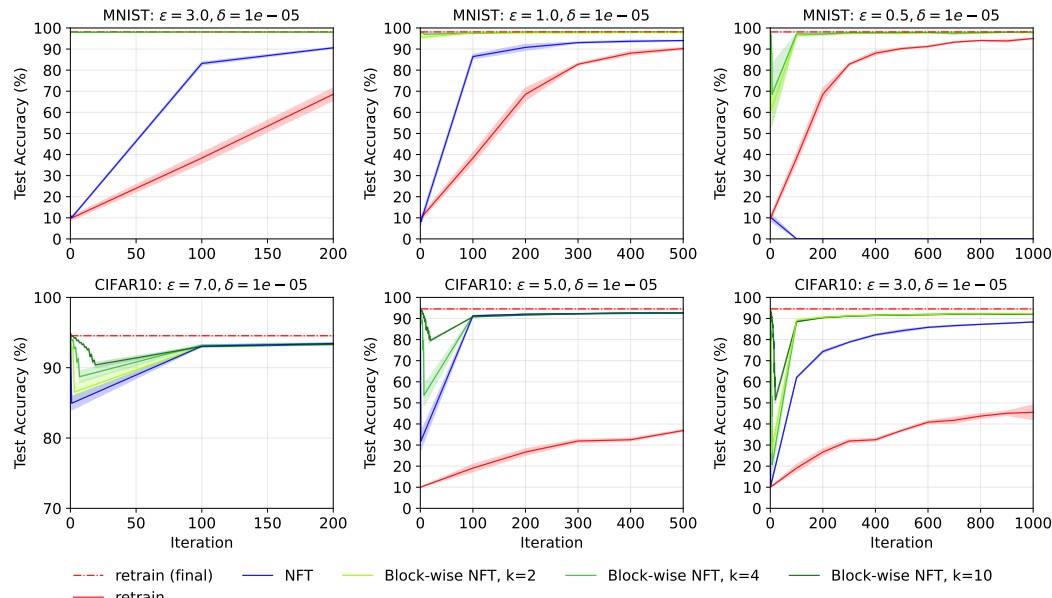

Figure 3: **Random 10% deletion on MNIST and CIFAR-10.** We compare standard Noisy Fine-Tuning (NFT) with Block-wise NFT (k=2,4,10) with the final retrain accuracy shown for reference. Across privacy budgets Block-wise NFT shows smoother, more stable unlearning and better post–fine-tuning recovery; increasing $k$ further reduces early accuracy loss.

Table 1: **Class 5 deletion on CIFAR-10.** Reported metrics: UA, RA, TA, MIA, and RTE. Block-Wise NFT matches retraining on UA and MIA, while remaining competitive on RA and TA. Baseline results are taken from the official SaLUN repository (Fan et al., 2024).

| Method | UA | RA | TA | MIA | RTE |
|---|---|---|---|---|---|
| Retrain | 100.00 | 100.00 | 86.14 | 100 | 46.37 |
| FT | 47.43 | 99.96 | 95.57 | 47.4 | 2.6 |
| GA | 94.09 | 92.2 | 87.03 | 94.09 | 0.15 |
| IU | 98.98 | 98.18 | 93.42 | 98.98 | 0.5 |
| SalUN | 100 | 99.81 | 95.1 | 100 | 2.76 |
| $\ell_1$-sparse | 100 | 91.79 | 89.08 | 100 | 2.55 |
| Block-wise NFT | 100 | 96.18 | 83.37 | 100 | 0.85 |

# 6 CONCLUSION

We studied the limitations of perturbation-based certified unlearning methods. In particular, we showed that, under natural assumptions on the loss, the standard clipping-based analysis of NFT is overly conservative: it effectively rules out the possibility of maintaining accuracy while being faster than full retraining. This motivates our reformulation in terms of the closeness parameter $\Delta(\rho)$, which captures the practically relevant distance between fully trained and retrained models. Building on this insight, we proposed block-wise noisy fine-tuning, which empirically reduces the accuracy drop observed during unlearning. Even under our assumptions, the method retains strong Membership Inference Attack (MIA) protection, highlighting its practical relevance. In summary, our approach preserves formal certified unlearning guarantees while substantially mitigating utility loss, opening the door to practical and scalable certified unlearning methods.

Our findings suggest that current definitions of certified unlearning may be too strict: they enforce indistinguishability in ways that are not always aligned with practical goals, while still not guaranteeing model utility. A promising direction is to revisit these definitions, aiming for frameworks that both formalize "forgetting" more faithfully and better capture utility preservation. Beyond this conceptual aspect, further work includes exploring adaptive block decompositions, scaling to larger and more complex architectures, and narrowing the remaining gap between certified unlearning and retraining.

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

## A RELATED WORK

**Certified Machine Unlearning.** Certified machine unlearning has emerged as a practical alternative to full retraining, offering rigorous data-removal guarantees at much lower cost. Classical approaches estimate the retrained solution via single–step Newton and influence surrogates (Guo et al. (2020); Sekhari et al. (2021); Zhang et al. (2024)), or use projected/perturbed gradient methods (Neel et al. (2021); Chien et al. (2024)), coupled with randomized mechanisms to ensure statistical indistinguishability.

Many of these certificates are explicitly *DP-inspired*, applying noisy updates and privacy amplification ideas from differential privacy Dwork & Roth (2014); Balle et al. (2020). However, recent methods typically rely on strong assumptions (e.g., smoothness/strong convexity, knowledge of Hessian eigenvalues, or even a unique minimizer), which limit applicability to modern deep networks Zhang et al. (2024); Liu et al. (2023); Allouah et al. (2025). In this work we remain within the certified DP line, but we specifically adapt the noisy fine-tuning framework of Koloskova et al. (2025) to preserve certificates *without* convexity or uniqueness assumptions while mitigating the accuracy loss usually observed in DP-based unlearning.

**Empirical unlearning in Image Classification.** Empirical (approximate) unlearning methods aim to remove the influence of the forget set without formal certificates and are therefore evaluated using strong auditing attacks alongside simple utility/efficiency metrics. Representative methods include sparsity-driven approaches (Fan et al., 2024; Jia et al., 2023), fine-tuning on the retain set (Warnecke et al., 2023a), gradient ascent on the forget set (Thudi et al., 2022), and adversarial-example-based unlearning (Ebrahimpour-Boroojeny et al., 2025), among others.

Efficacy is typically assessed with unlearning-specific membership-inference attacks (MIAs) (Kurmanji et al., 2023; Jia et al., 2023; Ebrahimpour-Boroojeny et al., 2025), together with standard model-performance metrics. Although these methods provide no formal guarantees, they constitute strong practical baselines and useful evaluation tools for auditing certified approaches.

## B CODE AVAILABILITY

We release our implementation at `https://github.com/unlearn-blocks-2025/block-wise-noisy-fine-tuning`. The repository includes the code needed to reproduce our experiments.

## C   NOTATIONS

Table 2: Notation used throughout the paper

| Symbol | Description |
|---|---|
| $\mathcal{A}$ | Learning algorithm mapping a dataset to model parameters |
| $\mathcal{D}$ | Training dataset |
| $d$ | Dimension of the parameter (weight) space $\mathbb{R}^d$ |
| $\hat{\mathbf{x}}$ | Model parameters obtained from $\mathcal{A}(\mathcal{D})$ |
| $\mathcal{D}_f$ | Subset of $\mathcal{D}$ to be forgotten |
| $\mathcal{D}_r$ | Retained dataset, $\mathcal{D} \setminus \mathcal{D}_f$ |
| $\mathcal{U}$ | Unlearning mechanism updating parameters after deletion request |
| $\tilde{\mathbf{x}}$ | Model parameters output by $\mathcal{U}$ |
| $\bar{\mathcal{A}}$ | Certifying algorithm in the definition of $(\varepsilon, \delta)$-unlearning |
| $\varepsilon$ | Privacy/unlearning parameter controlling multiplicative slack |
| $\delta$ | Privacy/unlearning parameter controlling additive slack |
| $\mathbf{x}_0$ | Model parameters after initial clipping of $\hat{\mathbf{x}}$ with radius $C_0$ |
| $\mathbf{x}_t$ | Model parameters at iteration $t$ of noisy fine-tuning |
| $g_t$ | Gradient at step $t$ computed on the retained data $\mathcal{D}_r$ |
| $\boldsymbol{\xi}_t$ | Gaussian noise $\sim \mathcal{N}(0, \sigma^2 I_d)$ added at step $t$ |
| $\Pi_C$ | Clipping operator with radius $C$, $\Pi_C(\mathbf{v}) = \mathbf{v} \cdot \min\{\frac{C}{\|\mathbf{v}\|}, 1\}$ |
| $\hat{\mathbf{x}}'$ | Model parameters obtained by training on the retained dataset $\mathcal{D}_r$ |
| $\mathbf{x}'_t$ | Iterates produced by unlearning updates when initialized at $\hat{\mathbf{x}}'$ |
| $\sigma^2$ | Variance of the Gaussian noise in noisy fine-tuning |
| $T_{\text{retrain}}$ | Minimal number of retraining steps required to reach good accuracy on $\mathcal{D}_r$ |
| $T$ | Number of unlearning steps in NFT |
| $\alpha$ | Target accuracy level in the thought experiment |
| $p$ | Probability of achieving accuracy at least $\alpha$ after $T$ steps |
| $\mathbf{x}_{\text{init}}$ | Randomly initialized model parameters |
| $\Delta(\rho)$ | High-probability bound on the distance between $\hat{\mathbf{x}}$ and $\hat{\mathbf{x}}'$ |
| $\rho$ | Failure probability in the definition of $\Delta(\rho)$ |
| $q$ | Order of Rényi Differential Privacy |
| $\varepsilon^{\text{rényi}}$ | Privacy loss at order $q$ in Rényi DP |
| $\sigma^2$ | Variance of Gaussian perturbations in noisy fine-tuning |
| $T(\sigma^2)$ | Minimal number of unlearning steps required for a given noise variance $\sigma^2$ |
| $\sigma^2_{\min}$ | Minimal feasible noise variance ensuring $(\varepsilon, \delta)$-unlearning |
| $k$ | Number of subspaces (blocks) in the partition of $\mathbb{R}^d$ |
| $r_i$ | Dimension of the $i$-th subspace, with $\sum_{i=1}^k r_i = d$ |
| $A_i \in \mathbb{R}^{d \times r_i}$ | Matrix with orthonormal columns spanning subspace $V_i$ |
| $A$ | Concatenation $[A_1 \cdots A_k] \in \mathbb{R}^{d \times d}$ with $A^\top A = I_d$ |
| $B_i$ | Coordinate vector in $\mathbb{R}^{r_i}$ corresponding to $V_i$ in the decomposition of $W$ |
| $\mathcal{G}(W, W')$ | Decomposition gap between two weight vectors |
| $z^{(1)} \preceq z^{(2)}$ | Coordinate-wise inequality between two vectors in $\mathbb{R}^k$ |
| $W_d(\mu, \mu')$ | Decomposed Wasserstein distance between distributions $\mu, \mu'$ |
| $D_q^{(z)}(\mu \,\|\, \nu)$ | Decomposed shifted Rényi divergence with shift vector $z$ and order $q$ |
| $e_i$ | $i$-th standard basis vector in $\mathbb{R}^k$ |

## D   PROOFS

*Proof of Proposition 1.* Since $A = [A_1 \;\; \cdots \;\; A_k]$ is orthogonal, we have $A^\top A = I_d$. For any $W \in \mathbb{R}^d$ set $B = A^\top W$ and partition it into blocks $B^\top = [B_1^\top, \ldots, B_k^\top]$. Then

$$AB = AA^\top W = W,$$

which expands to $W = \sum_{i=1}^{k} A_i B_i$. Uniqueness follows from orthogonality: if $\sum_i A_i B_i = \sum_i A_i B_i'$, then $A^\top (\sum_i A_i (B_i - B_i')) = \sum_i B_i - B_i' = 0$, hence $B_i = B_i'$.   □

## D.1   PROOF OF THEOREM 1

We first recall the shifted Rényi divergence (Feldman et al. (2018)):

**Definition 7** (Rényi divergence). *Let $q > 0$, $q \neq 1$. The $q$-Rényi divergence between two probability distributions $\mu$ and $\nu$ is defined as*

$$D_q(\mu \parallel \nu) := \frac{1}{q-1} \log \mathbb{E}_{X \sim \nu} \left( \frac{\mu(X)}{\nu(X)} \right)^q.$$

Let us consider the reasoning presented in Koloskova's paper. Their proof contains a minor indexing mismatch in the recursion expansion, which did not affect the final result for their asymptotics, but it will be important for us. We corrected the indices in the product $p_t$ accordingly.

Let us revisit the reasoning in Koloskova et al. (2025). In their proof, the transition from equation (23) to equation (24) involves a minor computational error in solving the recursion. While this did not affect their asymptotic conclusions, it becomes relevant for our analysis. We correct this step by adjusting the indices in the product $p_t$ accordingly. The corrected parts are highlighted in green.

**Theorem 3** (Koloskova A.9, fixed indices in recursion). *Let $T, q \geq 1$, $\gamma_0, \ldots, \gamma_{T-1} \geq 0$, $\sigma_0, \ldots, \sigma_{T-1} > 0$, $\lambda \geq 0$ and consider the two sequences $\{x_t\}_{0 \leq t \leq T}$, $\{x_t'\}_{0 \leq t \leq T}$ as defined above. Denote by $D_q$ the Rényi divergence of order $q$. Assume that for every $t \in \{0, \ldots, T-1\}$, $\gamma_t \lambda < 1$. Denote for every $t \in \{0, \ldots, T-1\}$,*

$$s_t := 2\gamma_t C_1, \qquad \rho_t := 1 - \gamma_t \lambda. \tag{10}$$

*If there exist $a_0, \ldots, a_{T-1} \in \mathbb{R}_{\geq 0}$ such that*

$$\sum_{t=0}^{T-1} \left( \prod_{k=t+1}^{T-1} \rho_k \right) a_t = \left( \prod_{t=0}^{T-1} \rho_t \right) 2C_0 + \sum_{t=0}^{T-1} \left( \prod_{k=t+1}^{T-1} \rho_k \right) s_t, \tag{11}$$

*then*

$$D_q(x_T \parallel x_T') \leq \sum_{t=0}^{T-1} \frac{q a_t^2}{2\sigma_t^2}. \tag{12}$$

Let us attempt to find the optimal parameters. We will search for a solution under the following constraints:

- Suppose there exists a maximum allowable noise level at any given step, $\sigma \geq \sigma_i$.
- Suppose the weight decay $\lambda$ and learning rate $\gamma$ are the same for all steps.
- We will also seek a solution for fixed $C_0$, $C_1$, and RDP-budget $(\varepsilon^{\text{rényi}}, q)$.

For simplicity, let us denote $\alpha_t = \prod_{k=t+1}^{T-1} \rho_k$, then the condition on $a$ can be rewritten as

$$\sum_{t=0}^{T-1} \alpha_t a_t = \rho_0 \alpha_0 2C_0 + \sum_{t=0}^{T-1} \alpha_t s_t, \tag{13}$$

In fact, determining $T$ is equivalent to finding the first moment when the left-hand side exceeds the right-hand side:

$$\sum_{t=0}^{T-1} \alpha_t a_t \geq \rho_0 \alpha_0 2C_0 + \sum_{t=0}^{T-1} \alpha_t s_t. \tag{14}$$

Moreover, the unlearning budget $\varepsilon$ is computed directly from the constraint on the Rényi divergence:

$$D_q(x_T \parallel x_T') \leq \sum_{t=0}^{T-1} \frac{q a_t^2}{2\sigma_t^2} \leq \varepsilon^{\text{rényi}}. \tag{15}$$

Let $y$ be the vector with coordinates $y_i = \frac{a_i}{\sigma_i}$. By the constraint, we are searching for a solution such that

$$\|y\|_2^2 \ \leq \ \frac{2\varepsilon^{\text{rényi}}}{q}. \tag{16}$$

Observe that by proportionally increasing $a_i$ until the inequality becomes an equality, we do not increase the minimal number of epochs $T$ required for condition equation 14. Hence, at the optimum we may assume

$$\|y\|_2 = \sqrt{\frac{2\varepsilon^{\text{rényi}}}{q}}. \tag{17}$$

Next, consider the optimal choice of $a_i$ for our condition. By the Cauchy–Schwarz inequality,

$$\sum_{t=0}^{T-1} \alpha_t a_t \ \leq \ \sum_{t=0}^{T-1} (\alpha_t \sigma_t) \left(\frac{a_t}{\sigma_t}\right) \ \leq \ \|\alpha\sigma\| \cdot \|y\| \ \leq \ \sqrt{\frac{2\varepsilon^{\text{rényi}}}{q}} \sqrt{\sum_{i=0}^{T-1} \alpha_i^2 \sigma_i^2}. \tag{18}$$

Equality holds when the two vectors are proportional, which determines the optimal values of $a_i$.

Rewriting inequality equation 14, we obtain

$$\rho_0 \alpha_0 2 C_0 + \sum_{t=0}^{T-1} \alpha_t s_t \ \leq \ \sqrt{\frac{2\varepsilon^{\text{rényi}}}{q}} \sqrt{\sum_{i=0}^{T-1} \alpha_i^2 \sigma_i^2} \ \leq \ \sqrt{\frac{2\varepsilon^{\text{rényi}}\sigma^2}{q}} \sqrt{\sum_{i=0}^{T-1} \alpha_i^2}. \tag{19}$$

Thus, $T$ does not decrease if we set $\sigma_i = \sigma$ for every step.

For convenience, define

$$C_b := \sqrt{\frac{2\varepsilon^{\text{rényi}}\sigma^2}{q}}. \tag{20}$$

We now expand the inequality we aim to obtain, using the fact that

$$\alpha_t = \prod_{k=t+1}^{T-1} \rho_k = (1-\gamma\lambda)^{T-1-t}. \tag{21}$$

The left-hand side is

$$2C_0(1-\gamma\lambda)^T + 2\gamma C_1 \sum_{t=0}^{T-1}(1-\gamma\lambda)^{T-1-t}$$
$$= 2C_0(1-\gamma\lambda)^T + 2\gamma C_1 \frac{1-(1-\gamma\lambda)^T}{\gamma\lambda}, \tag{22}$$

while the right-hand side is

$$C_b\sqrt{\sum_{t=0}^{T-1}(1-\gamma\lambda)^{2T-2-2t}} \ = \ C_b\sqrt{\frac{1-(1-\gamma\lambda)^{2T}}{1-(1-\gamma\lambda)^2}}. \tag{23}$$

Observe that both sides of the inequality are positive, hence it suffices to require that the square of the left-hand side exceeds the square of the right-hand side.

Introduce the variable

$$x = (1-\gamma\lambda)^T. \tag{24}$$

Then $T$ can be recovered from $x$ as

$$T = \frac{\log(x)}{\log(1-\gamma\lambda)}. \tag{25}$$

Thus, the problem of finding the minimal $T$ is equivalent to finding the *maximal* $x$, subject to the constraint $0 < x \leq 1$.

By rewriting the difference of squares of the left- and right-hand sides, the problem reduces to finding the maximal root in the half-interval $(0, 1]$ of the equation

$$\left(2C_0 x + 2\gamma C_1 \frac{1-x}{\gamma\lambda}\right)^2 - C_b^2 \cdot \frac{1-x^2}{1-(1-\gamma\lambda)^2} = 0. \tag{26}$$

Equivalently,

$$\left(\left(\tfrac{2C_0}{C_b} - \tfrac{2C_1}{\lambda C_b}\right)x + \tfrac{2C_1}{\lambda C_b}\right)^2 - \tfrac{1-x^2}{1-(1-\gamma\lambda)^2} = 0. \tag{27}$$

Define

$$\zeta := \tfrac{1}{1-(1-\gamma\lambda)^2}, \qquad \beta_0 := \tfrac{2C_1}{\lambda C_b}, \qquad \beta_1 := \tfrac{2C_0}{C_b} - \tfrac{2C_1}{\lambda C_b} = \beta_0\left(\tfrac{\lambda C_0}{C_1} - 1\right). \tag{28}$$

Then the quadratic equation can be written as

$$(\beta_1^2 + \zeta)x^2 + 2\beta_0\beta_1 x + (\beta_0^2 - \zeta) = 0. \tag{29}$$

Its discriminant is

$$\begin{aligned}\Delta &= 4\beta_0^2\beta_1^2 - 4(\beta_1^2 + \zeta)(\beta_0^2 - \zeta) \\ &= 4\zeta(\zeta + \beta_1^2 - \beta_0^2).\end{aligned} \tag{30}$$

The largest root of the quadratic is given by

$$x_{\max} = \frac{-2\beta_0\beta_1 + \sqrt{4\zeta(\zeta + \beta_1^2 - \beta_0^2)}}{2(\beta_1^2 + \zeta)}. \tag{31}$$

We now state the condition for the existence of a root:

$$\zeta \geq \beta_0^2 - \beta_1^2 = \beta_0^2 - \left(\tfrac{2C_0}{C_b} - \beta_0\right)^2 = \tfrac{8C_0C_1}{C_b^2\lambda} - \tfrac{4C_0^2}{C_b^2}. \tag{32}$$

Equivalently,

$$\tfrac{1}{1-(1-\gamma\lambda)^2} \geq \tfrac{8C_0C_1}{C_b^2\lambda} - \tfrac{4C_0^2}{C_b^2} = \tfrac{2q}{\sigma^2 \varepsilon^{\text{rényi}}}\left(\tfrac{2}{\lambda}C_0C_1 - C_0^2\right). \tag{33}$$

This implies

$$\sigma^2 \geq \gamma(2 - \gamma\lambda) \cdot \tfrac{2q}{\varepsilon^{\text{rényi}}} \cdot \left(2 - \tfrac{\lambda C_0}{C_1}\right) \cdot C_0C_1, \tag{34}$$

In the case $\frac{\lambda C_0}{C_1} \in (0, 1)$, the right-hand side of the inequality is positive, which yields a bound on the minimal $\sigma$ in this regime.

To obtain the bound on $T$, it suffices to substitute all original variables into the formula 31 for $x_{\max}$.

In the case of the minimal $\sigma$, the discriminant vanishes, and the expression simplifies to

$$\begin{aligned}x_{\max} &= \frac{-\beta_0\beta_1}{\beta_1^2 + \zeta} = \frac{-\beta_0\beta_1}{\beta_0^2} = \frac{-(2\lambda C_0 - 2C_1)}{2C_1} \\ &= 1 - \tfrac{\lambda C_0}{C_1} \in (0, 1].\end{aligned} \tag{35}$$

Therefore, substituting into

$$T = \tfrac{\log(x)}{\log(1-\gamma\lambda)}, \tag{36}$$

we obtain the desired estimate for $T$.

Furthermore, for $\frac{\lambda C_0}{C_1} \in [1, 2)$, the bound on the noise obtained from the discriminant is also positive; however, in this case, the corresponding $x_{\max}$ would fall outside the interval $(0, 1)$, as required.

To analyze the regime $\frac{\lambda C_0}{C_1} \geq 1$, we observe that $\beta_1 \leq 0$, implying that the vertex of the parabola lies at a negative coordinate. Consequently, the value at $0$ must be negative (since the value at $1$ is positive). From the inequality $\zeta > \beta_0^2$, one therefore derives a bound on $\sigma^2$. It is worth noting, however, that in the case of equality, $x_{\max}$ becomes zero, which corresponds to an infinite $T$.

**Formula for minimal $\sigma$ with given $T$**    Given argumentation above, we can express $\sigma$ in terms of $x_{\max}$:

$$\text{Let} \quad z := 1 - \frac{\lambda C_0}{C_1}, \qquad x := x_{\max} \in (0, 1]. \tag{37}$$

We now rewrite the formula 31 for $x_{\max}$ in terms of $\sigma$, which leads to a quadratic equation in $s := 1/\sigma^2$:

$$as^2 + bs + c = 0, \tag{38}$$

where

$$a = \left( \frac{2q\,C_1^2}{\varepsilon^{\text{rényi}}\,\lambda^2} \right)^2 z^2 \, (1 - xz)^2,$$

$$b = \left( \frac{2q\,C_1^2}{\varepsilon^{\text{rényi}}\,\lambda^2} \right) \frac{1}{\gamma\lambda(2 - \gamma\lambda)} \Big[ 1 - 2xz + (2x^2 - 1)z^2 \Big], \tag{39}$$

$$c = \frac{x^2 - 1}{\big( \gamma\lambda(2 - \gamma\lambda) \big)^2}.$$

Observe that $a > 0$ and $c < 0$, hence we are interested in the largest root of the quadratic.

The (positive) solution for $\sigma^2$ can then be written as

$$\sigma^2(x) = \frac{2a}{-b + \sqrt{b^2 - 4ac}}. \tag{40}$$

Or the equivalent expression

$$\sigma^2(x) = \frac{-b - \sqrt{b^2 - 4ac}}{2c}. \tag{41}$$

### D.2   PROOF OF THE THEOREM 2

The proof proceeds analogously to that of the original theorem (Koloskova et al., 2025).

We adapt Lemma A7 from the original proof, using adapted definitions of $\infty$-Wasserstein distance and shifted Rényi divergence.

**Lemma 2** (Decomposed Lemma A7 (Koloskova et al., 2025)))**.** *Let $q \geq 1$, $z, \rho, s \geq 0$, $\psi : \mathbb{R}^d \to \mathbb{R}^d$, and $X, Y$ be arbitrary random variables. If $\psi$ satisfies, for all $\mathbf{x}, \mathbf{x}' \in \mathbb{R}^d$ (for a single component $i$, while for the others nothing changes),*

$$\|\psi(\mathbf{x}') - \psi(\mathbf{x})\| \; \leq \; \rho\|\mathbf{x}' - \mathbf{x}\| + s,$$

*then*

$$D_q^{(z_0, \ldots, (\rho z_i + s), \ldots, z_k)}(\psi(X) \,\|\, \psi(Y)) \; \leq \; D_q^{(z_0, \ldots, z_i, \ldots, z_k)}(X \,\|\, Y).$$

*Proof.* To adapt the original proof, it suffices to observe that for the decomposed Wasserstein distance the following inequality holds:

from $W_d(\mu, \nu) \preceq z = (0, \ldots, z_i, \ldots, 0)$

$$W_d(\psi_{\#}(\mu), \psi_{\#}(\nu)) \preceq (0, \ldots, (pz_i + s), \ldots, 0),$$

when we modify the $i$-th component. This inequality indeed holds due to the assumption on $\psi$: for every pair $(x, x')$, the condition ensures the bound componentwise.    $\square$

Using the lemma above and decomposed Shift Reduction Lemma (1), we can proceed block by block. By zeroing out the coordinates sequentially, we bound the *increment* in Rényi divergence contributed by block $i$ by a quantity $\varepsilon_i^{\text{rényi}}$. Summing these contributions yields an overall

$$(\textstyle\sum_{i=1}^k \varepsilon_i^{\text{rényi}},\, q)\text{-RDP}.$$

Applying the standard conversion from Rényi DP to $(\varepsilon, \delta)$-DP (Mironov, 2017) gives

Since we already have an estimate for the number of steps required for unlearning in each component, summing them gives the bound on the total number of steps.

$$\varepsilon(\delta) \;=\; \sum_{i=1}^{k} \varepsilon_i^{\text{rényi}} \;+\; \frac{\log(1/\delta)}{q-1} \;=\; \sum_{i=1}^{k} \varepsilon_i \;-\; (k-1)\frac{\log(1/\delta)}{q-1}, \tag{42}$$

where we define $\varepsilon_i := \varepsilon_i^{\text{rényi}} + \frac{\log(1/\delta)}{q-1}$ for notational convenience.

*We emphasize that we do not claim each block is itself an $(\varepsilon_i, \delta)$-mechanism; the equality above is an algebraic rewriting of the single $(\sum_i \varepsilon_i^{\text{rényi}}, q)$-RDP bound after conversion.*

**Proof of corollary 1** To guarantee the same budget as the baseline algorithm, it suffices to set

$$\varepsilon_i^{\text{renyi}} = \frac{\varepsilon^{\text{renyi}}}{k}.$$

Moreover, due to the randomness in the distribution of the model weights and gradients, the group norms are approximately equal. Hence we may treat

$$\frac{1}{\sqrt{k}}C_0 \quad \text{and} \quad \frac{1}{\sqrt{k}}C_1$$

as the individual clipping bounds for $B_i$.

Keeping the same noise level (not necessarily minimal) does not change the number of steps $T$ for each group, since the factors of $\sqrt{k}$ compensate.

Thus, the total cost is exactly $kT$ steps for the whole algorithm.

Finally, we may choose a larger noise level, thereby reducing $T$. Since in each step we add substantially less noise to the model than in the baseline algorithm, we have the flexibility to increase the noise.

# E EXPERIMENTS

## E.1 PARAMETERS AND IMPLEMENTATION DETAILS

Listing 1: Architecture of the model used for MNIST.

```python
import torch.nn as nn
import torch.nn.functional as F

class LinearNet(nn.Module):
    def __init__(self, num_classes: int = 10):
        super().__init__()
        self.flatten = nn.Flatten()
        self.fc1 = nn.Linear(28 * 28, 2048)
        self.fc2 = nn.Linear(2048, 1024)
        self.fc3 = nn.Linear(1024, 512)
        self.fc4 = nn.Linear(512, 256)
        self.fc5 = nn.Linear(256, num_classes)

    def forward(self, x):
        x = self.flatten(x)
        x = F.relu(self.fc1(x))
        x = F.relu(self.fc2(x))
        x = F.relu(self.fc3(x))
        x = F.relu(self.fc4(x))
        x = self.fc5(x)
        return x
```

Table 3: *Training* hyperparameters for the fully trained MNIST model and the retrain baseline (identical settings).

| Parameter | Fully trained | Retrain baseline |
|---|---|---|
| Optimizer | SGD | SGD |
| Learning rate | 0.01 | 0.01 |
| Momentum | 0.9 | 0.9 |
| Weight decay | $1 \times 10^{-5}$ | $1 \times 10^{-5}$ |
| Batch size | 64 | 64 |

Table 4: Training hyperparameters for the fully trained CIFAR-10 model and the retrain baseline (identical settings).

| Parameter | Fully trained | Retrain baseline |
|---|---|---|
| Optimizer | SGD | SGD |
| Learning rate | 0.1 | 0.1 |
| Momentum | 0.9 | 0.9 |
| Weight decay | $5 \times 10^{-4}$ | $5 \times 10^{-4}$ |
| Batch size | 256 | 256 |
| LR scheduler | MultiStep(91, 136) $\times 0.1$; 182 epochs | same |
| Data augmentation | Crop(32,pad=4)+Flip+Norm($\mu, \sigma$) | same |

Table 5: Unlearning hyperparameters used in experiments on MNIST.

| Parameter | Random 10% deletion | Classwise deletion |
|---|---|---|
| Initial distance bound $\Delta(\rho)$ | 0.01 | 0.02 |
| Unlearning step size $\eta$ | $1 \times 10^{-4}$ | $1 \times 10^{-3}$ |
| Weight decay $\lambda$ (unlearning) | 10 | 30 |
| Total privacy budget $\varepsilon$ | 1.0 | 3.0 |
| Failure probability $\delta$ | $10^{-5}$ | $10^{-5}$ |
| Optimal RDP order $q_{\mathrm{opt}}$ | 24.50 | 9.10 |
| Sum over blocks of Rényi $\varepsilon_{\mathrm{rényi}}$ | 0.510 | 1.579 |
| number of steps $T$ (per block) | 2 | 2 |
| noise variance $\sigma^2$ | 0.0980 | 0.1505 |
| Fine-tuning learning rate | $1 \times 10^{-2}$ | $1 \times 10^{-2}$ |
| Fine-tuning weight decay | $1 \times 10^{-5}$ | $1 \times 10^{-5}$ |
| Fine-tuning momentum | 0.9 | 0.9 |

Table 6: Per-block scaling across the number of blocks $k$ on MNIST.

| | Random 10% deletion | | Classwise deletion | |
|---|---|---|---|---|
| $k$ | $C_1/\sqrt{k}$ | $\varepsilon^{\mathrm{rényi}}/k$ | $C_1/\sqrt{k}$ | $\varepsilon^{\mathrm{rényi}}/k$ |
| 2 | 70.71 | 0.255 | 70.71 | 0.789 |
| 4 | 50.00 | 0.128 | 50.00 | 0.395 |
| 7 | 37.80 | 0.073 | 37.80 | 0.226 |
| 10 | 31.62 | 0.051 | 31.62 | 0.158 |
| 13 | 27.74 | 0.039 | 27.74 | 0.121 |

## E.2 EXPERIMENTS WITH DIFFERENT BLOCK STRUCTURE

We further compare three constructions of the block mixing matrix $A$ (see Section 4.3):

- a random $A$ with equal-sized blocks;

- a permutation-matrix $A$ with equal-sized blocks;

- a cyclic layer-wise grouping into $k$ blocks, assigning layer $\ell$ to block $\ell \bmod k$

- a two-block partition on head and the rest of the model (only for resnet-18).

We provide results of experiments for both dataset (MNIST and CIFAR-10). For forget set $\mathcal{D}_f$ we are using random 10% deletion task.

Table 7: Unlearning hyperparameters used in experiments on CIFAR$-10$.

| Parameter | Random 10% deletion | Classwise deletion |
|---|---|---|
| Initial distance bound $\Delta(\rho)$ | 0.01 | 0.05 |
| Unlearning step size $\eta$ | $1 \times 10^{-4}$ | $1 \times 10^{-3}$ |
| Weight decay $\lambda$ (unlearning) | 30 | 3 |
| Total privacy budget $\varepsilon$ | 5 | 10 |
| Failure probability $\delta$ | $10^{-5}$ | $10^{-3}$ |
| Optimal RDP order $q_{\mathrm{opt}}$ | 6.06 | 2.77 |
| Sum over blocks of Rényi $\varepsilon_{\mathrm{rényi}}$ | 2.725 | 6.101 |
| number of steps $T$ (per block) | 2 | 2 |
| noise variance $\sigma^2$ | 0.0178 | 0.0499 |
| Fine-tuning learning rate | $1 \times 10^{-3}$ | $1 \times 10^{-3}$ |
| Fine-tuning weight decay | $5 \times 10^{-4}$ | $5 \times 10^{-4}$ |
| Fine-tuning momentum | 0.9 | 0.9 |

Table 8: Per-block scaling across the number of blocks $k$ for CIFAR-10.

| | Random 10% deletion | | Classwise deletion | |
|---|---|---|---|---|
| $k$ | $C_1/\sqrt{k}$ | $\varepsilon^{\mathrm{rényi}}/k$ | $C_1/\sqrt{k}$ | $\varepsilon^{\mathrm{rényi}}/k$ |
| 2 | 38.891 | 1.363 | 38.891 | 3.051 |
| 4 | 27.500 | 0.681 | 27.500 | 1.525 |
| 7 | 20.788 | 0.389 | 20.788 | 0.872 |
| 10 | 17.393 | 0.273 | 17.393 | 0.610 |
| 13 | 15.254 | 0.210 | 15.254 | 0.469 |

**Results on MNIST** Figure 4 shows MNIST trajectories at privacy budgets $\varepsilon \in \{0.5, 1.0\}$, $\delta = 10^{-5}$ and number of blocks $k = 2$. The three schemes demonstrate comparable stability during unlearning and recovery under fine-tuning.

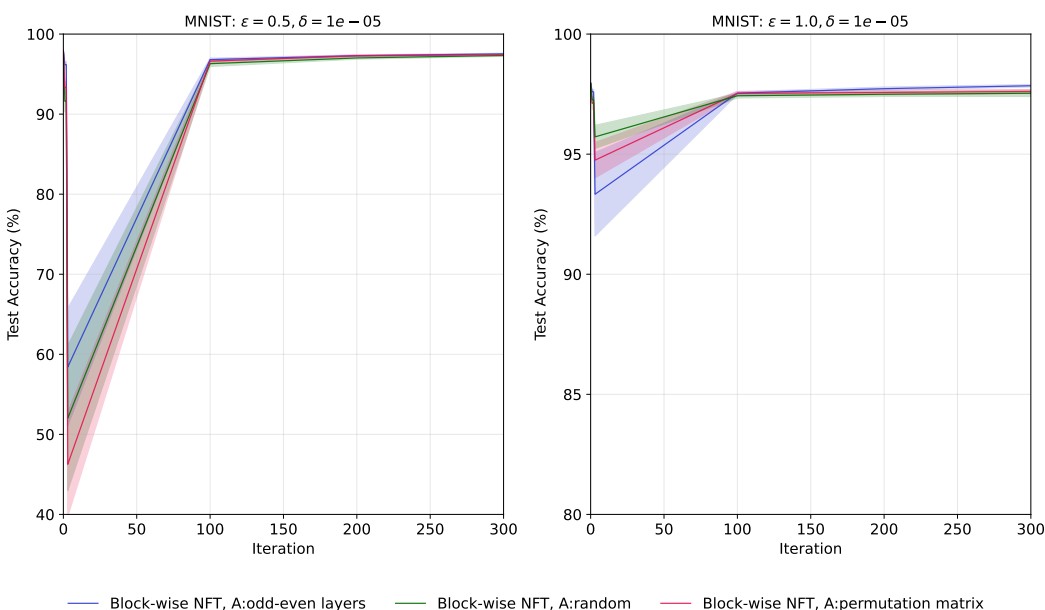

Figure 4: **Block-construction schemes** for Block-wise NFT on MNIST at $\varepsilon \in \{0.5, 1.0\}$ and $\delta = 10^{-5}$.

**Results on CIFAR-10** Figure 5 shows CIFAR-10 trajectories for privacy budget $\varepsilon = 5.0$ and $\delta = 10^{-5}$, with the number of blocks equals to 4 for the first 3 methods (partition head/body does not allow any other block number).

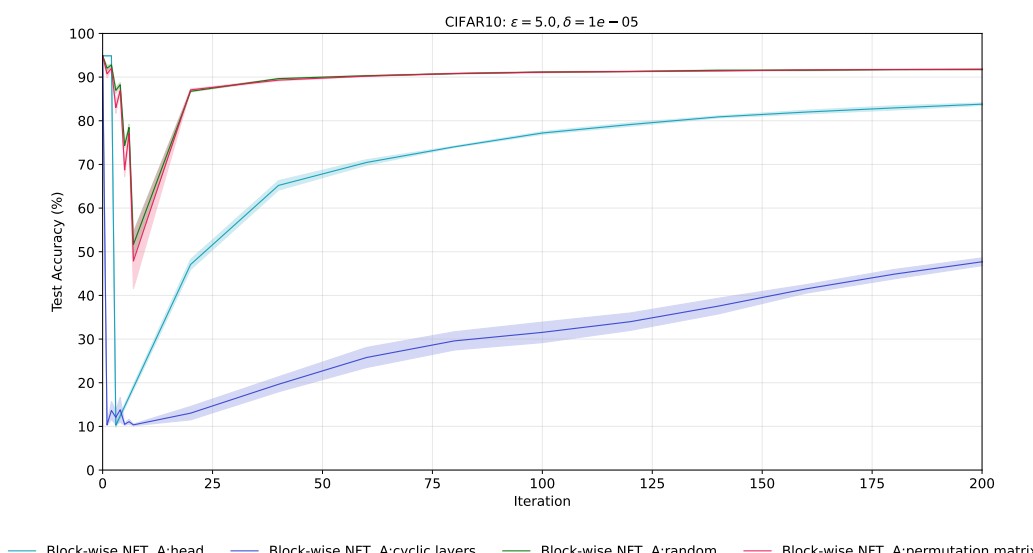

Figure 5: Block-construction schemes for Block-wise NFT on CIFAR−10 at $\varepsilon = 5.0$ and $\delta = 10^{-5}$.

### E.3  CLASS-WISE EXPERIMENTS

We evaluate class-wise forgetting on MNIST, where we forget class 5 entirely.

Table 9 reports UA/RA/TA/MIA for $\varepsilon \in \{3.0, 1.0\}$: Block-wise NFT attains UA=100 and MIA=100 at both budgets while preserving higher RA/TA than NFT.

Figure 6 shows the corresponding learning dynamics, with a smaller transient drop and clearer recovery under fine-tuning.

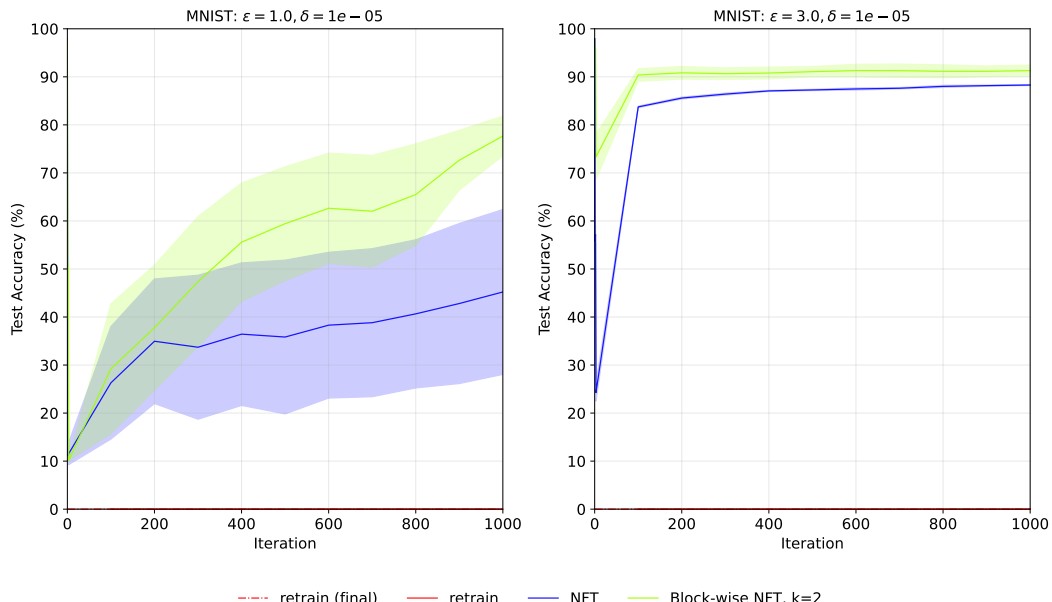

Figure 6: Caption about forget class

Table 9: **MNIST, class-wise forgetting.** Evaluating unlearning metrics for NFT and Block-wise NFT across $\varepsilon \in \{3.0, 1.0\}$. Block-wise NFT attains the best possible score for UA and MIA while preserving RA/TA better than NFT.

| Method | UA | RA | TA | MIA |
|--------|-----|-----|-----|------|
| Retrain | 100.00 | 99.83 | 89.33 | 100.00 |
| NFT $\varepsilon = 3.0$ | 100.00 | 97.84 | 88.23 | 100.00 |
| Block-wise NFT $\varepsilon = 3.0$ | 100 | 99.5 | 89.44 | 100.00 |
| NFT $\varepsilon = 1.0$ | 100.00 | 88.5 | 81.0 | 100.00 |
| Block-wise NFT $\varepsilon = 1.0$ | 100.00 | 92.65 | 84.37 | 100.00 |

### E.4 ADDITIONAL EXPERIMENTS

We also report results for a more challenging setup in which 50% of the training data are selected uniformly at random for removal.

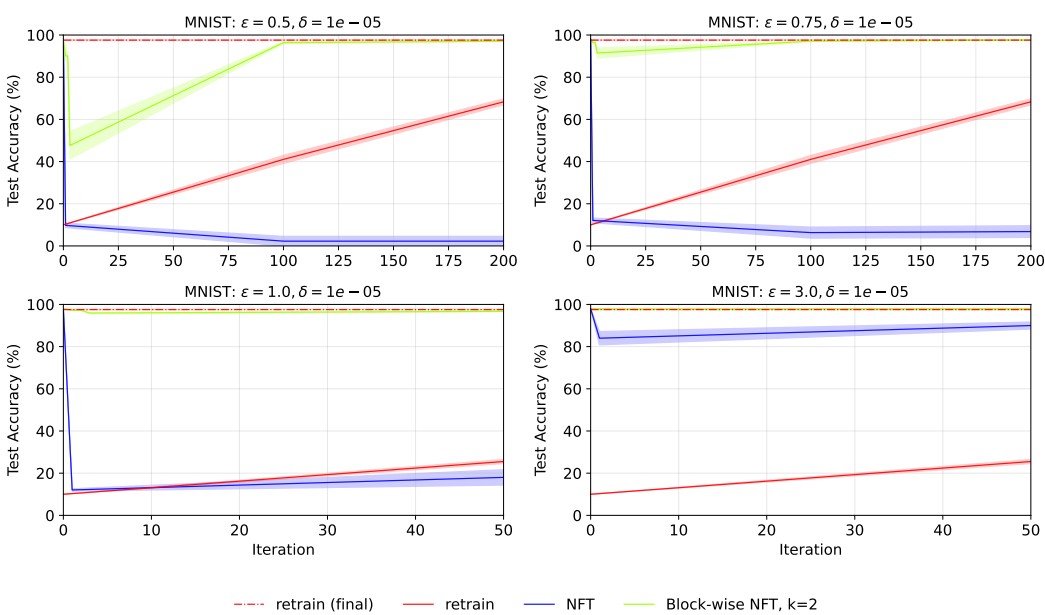

Figure 7: **Random 50% deletion.** We compare standard NFT with Block-wise NFT with two blocks, and show the final accuracy of retraining from scratch for reference. Despite the larger forget set, Block-wise NFT maintains greater stability (smaller initial drop, smoother curves) and achieves stronger recovery.

### E.5 EXTENDED EXPERIMENTS FOR CIFAR-10

In this section we provide the full per-class unlearning results for CIFAR-10. For each of the 10 classes, we delete the class entirely, retrain the baseline model from scratch on the retained data, and apply Block-wise NFT under the same setting. The results are presented in Table 10.

**Note on runtime.** The RTE values reported in Table 1 were obtained in a different runtime session than the experiments in the main paper, resulting in a slight systematic shift in wall-clock time. Since RTE is used only as a relative measure within each experiment, this does not affect any comparisons or conclusions.

Table 10: **Per-class deletion results on CIFAR-10.** For each deleted class, we report results for retraining-from-scratch and for Block-wise NFT (our method). Metrics: unlearned accuracy (UA), retain accuracy (RA), test accuracy (TA), membership-inference score (MIA), and relative training effort (RTE, minutes).

| | Retrain | | | | | Block-wise NFT | | | | |
|---|---|---|---|---|---|---|---|---|---|---|
| **Class** | UA | RA | TA | MIA | RTE | UA | RA | TA | MIA | RTE |
| **0** | 100.00 | 100.00 | 85.21 | 100.00 | 44.35 | 100.00 | 97.17 | 83.57 | 100.00 | 0.80 |
| **1** | 100.00 | 100.00 | 84.98 | 100.00 | 44.29 | 100.00 | 96.89 | 82.93 | 100.00 | 0.81 |
| **2** | 100.00 | 100.00 | 85.56 | 100.00 | 44.29 | 100.00 | 97.24 | 83.64 | 100.00 | 0.80 |
| **3** | 100.00 | 100.00 | 86.47 | 100.00 | 44.26 | 100.00 | 97.72 | 84.99 | 100.00 | 0.81 |
| **4** | 100.00 | 100.00 | 85.00 | 100.00 | 44.53 | 100.00 | 97.06 | 83.73 | 100.00 | 0.81 |
| **5** | 100.00 | 100.00 | 86.14 | 100.00 | 44.35 | 100.00 | 96.18 | 84.33 | 100.00 | 0.80 |
| **6** | 100.00 | 100.00 | 85.06 | 100.00 | 44.35 | 100.00 | 97.05 | 82.87 | 100.00 | 0.81 |
| **7** | 100.00 | 100.00 | 84.83 | 100.00 | 44.07 | 100.00 | 96.96 | 83.32 | 100.00 | 0.80 |
| **8** | 100.00 | 100.00 | 85.03 | 100.00 | 44.47 | 100.00 | 96.97 | 83.21 | 100.00 | 0.80 |
| **9** | 100.00 | 100.00 | 84.9 | 100.00 | 44.29 | 100.00 | 97.10 | 83.32 | 100.00 | 0.81 |
| **Mean** | 100.00 | 100.00 | 85.32 | 100.00 | 44.33 | 100.00 | 96.88 | 83.79 | 100.00 | 0.80 |
| **Std** | 0 | 0 | 0.53 | 0 | 0.11 | 0 | 0.47 | 0.63 | 0 | 0.01 |

### E.6 EXPERIMENTS ON VIT-TINY

To demonstrate that our method applies beyond convolutional architectures, we evaluate SSNI on a transformer model, VIT-TINY, trained on CIFAR-10. Following standard practice, we initialize all experiments from the *pretrained* ViT-Tiny checkpoint provided in the official implementation. The same pretrained model is used for retrain-from-scratch and fully-trained model to ensure fairness and comparability.

In this experiment, the forget set consists of **10% of the CIFAR-10 training data**, sampled uniformly at random across all classes. We report the *test accuracy* throughout unlearning and subsequent fine-tuning. Results are shown for two noise budgets, $\varepsilon = 5$ (left) and $\varepsilon = 7$ (right).

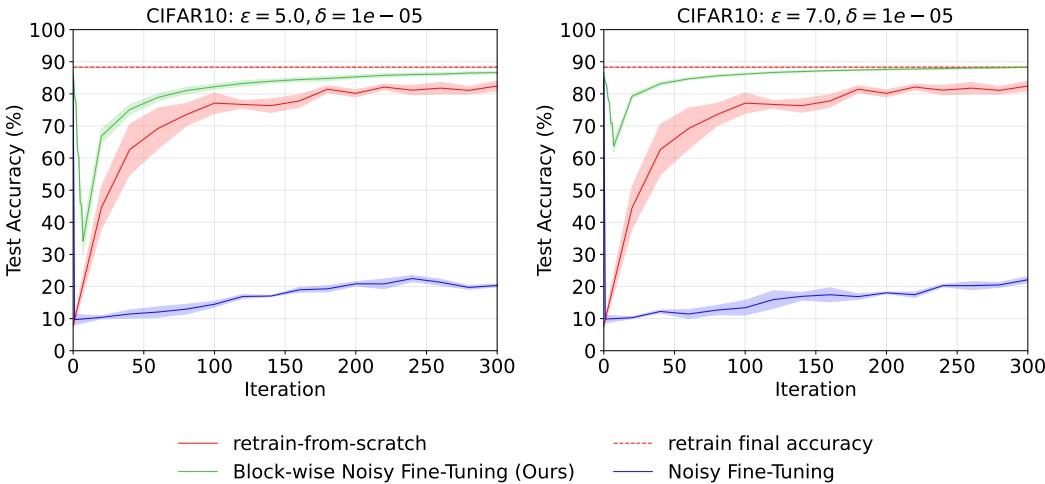

Figure 8: **ViT-Tiny unlearning on CIFAR-10 with 10% forget set.** Test accuracy vs. training steps for different noise budgets. NFT becomes unstable and consistently underperforms full retraining for both $\varepsilon = 5$ and $\varepsilon = 7$. Block-wise NFT significantly stabilizes training and stays much closer to the retrain curve and final retrain accuracy.

**Hyperparameters.** We use the standard ViT-Tiny architecture with classification head for 10 classes. The block decomposition is computed *once* before unlearning and reused for all iterations.

Table 11: Training hyperparameters for the ViT-Tiny model and the retrain baseline (identical settings). The model is instantiated via `timm.create_model("vit_tiny_patch16_224", pretrained=True)`.

| Parameter | Pretrained ViT-Tiny | Retrain baseline |
|---|---|---|
| Model architecture | ViT-Tiny (patch 16) | same |
| Pretrained initialization | Yes | Yes |
| Optimizer | AdamW | AdamW |
| Learning rate | $3 \times 10^{-4}$ | $3 \times 10^{-4}$ |
| Weight decay | 0.05 | 0.05 |
| Batch size | 64 | 64 |
| Epochs | 50 | 50 |
| LR scheduler | CosineAnnealingLR ($T_{\max} = 50$) | same |
| Warmup | none | none |
| Image resolution | 224 (ViT input) | 224 |
| Data augmentation | Resize(224) + RandomHorizontalFlip + ToTensor + Normalize($\mu, \sigma$) | same |

Table 12: Unlearning hyperparameters for Block-Wise NFT and NFT on ViT-Tiny for budgets $\varepsilon = 5$ and $\varepsilon = 7$. All Block-Wise NFT runs use $k = 4$ blocks.

| Parameter | Block-NFT/NFT ($\varepsilon = 5$) | Block-NFT/NFT ($\varepsilon = 7$) |
|---|---|---|
| Initial distance bound $\Delta(\rho)$ | 0.05 | 0.05 |
| Per-block clipping $C_1$ | 15 | 15 |
| Unlearning step size $\eta$ | 0.002 | 0.002 |
| Weight decay $\lambda$ (unlearning) | 25 | 25 |
| Total privacy budget $\varepsilon$ | 5 | 7 |
| Failure probability $\delta$ | $10^{-5}$ | $10^{-5}$ |
| number of steps $T$ (per block) | 2 | 2 |
| noise variance $\sigma^2$ | 0.079 | 0.058 |
| Fine-tuning optimizer | AdamW | AdamW |
| Fine-tuning learning rate | $5 \times 10^{-3}$ | $5 \times 10^{-3}$ |
| Fine-tuning weight decay | 0 | 0 |

Table 13: Per-block scaling for ViT-Tiny with $k = 4$ blocks. The clipping radius scales as $C_1/\sqrt{k}$ and the Rényi privacy budget scales as $\varepsilon_{\text{rényi}}/k$.

| Quantity | Value |
|---|---|
| Number of blocks $k$ | 4 |
| Per-block clipping $C_1/\sqrt{k}$ | 7.5 |
| Per-block Rényi budget $\varepsilon_{\text{rényi}}/k$ | 0.681 |

# F  BOUNDING THE PROXIMITY $\Delta(\rho)$: THEORETICAL AND EMPIRICAL PERSPECTIVES

The key quantity in our certificates is the discrepancy $\Delta(\rho)$ between the full retrain and the retraining-from-scratch. While obtaining a tight bound on $\Delta(\rho)$ for *arbitrary* deep models is challenging, this question is well-studied in the literature on *argument stability* (also known as parameter stability). Below we summarize regimes where such bounds are known or can be derived under standard assumptions.

## F.1  THEORETICAL BOUNDS VIA ARGUMENT STABILITY

Algorithmic stability has long been used to analyze how sensitive learning algorithms are to changes in the training data. Liu et al. (2017) formalized *argument stability*, which upper-bounds the parameter deviation between hypotheses trained on neighboring datasets. Several subsequent works derive explicit bounds on this deviation under standard assumptions.

**Smooth and Lipschitz losses.** Hardt et al. (2016) provide a trajectory-level recursion for SGD under $\beta$-smooth and $L$-Lipschitz losses in three regimes: (i) non-convex, (ii) convex, (iii) $\gamma$-strongly convex. When the replaced index is known — as in our full-vs-retained setting — their recursion

yields a closed-form upper bound on

$$\|\theta^{(t)} - \theta'^{(t)}\|,$$

and hence on our proximity $\Delta(\rho)$.

**Example (strongly convex case).** For completeness, we briefly illustrate how a bound on $\Delta(\rho)$ follows from the recursion of Hardt et al. (2016). Suppose the loss is $\gamma$-strongly convex and $\beta$-smooth, and SGD uses a constant stepsize $\alpha \leq 1/\beta$. Let $\theta^{(t)}$ and $\theta'^{(t)}$ denote the SGD iterates obtained from the full and retained datasets, respectively, and let $\delta_t = \|\theta^{(t)} - \theta'^{(t)}\|$. Since both runs start from the same initialization, we have $\delta_0 = 0$.

Hardt et al. (2016) give the following one-step inequalities:

1. If the minibatches coincide:
$$\delta_{t+1} \leq (1 - \alpha\gamma)\,\delta_t.$$

2. If the minibatches differ:
$$\delta_{t+1} \leq (1 - \alpha\gamma)\,\delta_t + 2\alpha L.$$

Unrolling this recursion yields the explicit bound

$$\delta_T \leq 2\alpha L \sum_{k \in B} (1 - \alpha\gamma)^k,$$

where $B$ is the set of iteration indices for which the minibatches differ. This provides a closed-form upper bound on $\delta_T$, and therefore on $\Delta(\rho)$ in this regime.

**Nonsmooth convex losses.** Bassily et al. (2020) prove argument stability for SGD without requiring smoothness. Using the monotonicity of subgradients together with $L$-Lipschitzness, they bound the deviation $\|\theta^{(t)} - \theta'^{(t)}\|$ at every iteration, directly giving an upper bound on $\Delta(\rho)$.

**Neural networks.** For certain neural architectures, stability of gradient-based methods has also been established. The work of Lei et al. (2022) proves stability-based generalization bounds for shallow ReLU networks. Complementarily, Richards & Kuzborskij (2021) analyze the dynamics of gradient descent and obtain stability and generalisation guarantees for shallow networks beyond the NTK regime. In both cases, the analysis controls how the learned parameters change under small perturbations of the training data, providing architecture-specific upper bounds on the parameter deviation and hence on $\Delta(\rho)$ for these models.

Together, these results show that $\Delta(\rho)$ can be bounded in several well-understood regimes — including smooth non-convex objectives, nonsmooth convex objectives, and specific neural-network architectures for which parameter-stability analyses exist. Establishing such bounds for arbitrary deep networks remains an open problem, but many practical settings already fall into one of the regimes above.

## F.2 PRACTICAL ESTIMATION AND CALIBRATION OF $\Delta(\rho)$

Even when a closed-form theoretical bound is not available, $\Delta(\rho)$ can still be calibrated empirically for a given architecture. A single calibration run estimates how far the model typically moves under small controlled dataset perturbations, and a conservative value of $\Delta(\rho)$ can then be fixed for future unlearning requests.

**Architectural sensitivity estimation.** To obtain such a calibration, we measure the parameter deviation induced by retraining on slightly perturbed datasets (e.g., replacing a small random subset). This procedure is performed once per architecture and yields a conservative upper estimate of $\Delta(\rho)$.

**Using $\Delta(\rho)$ as a calibration parameter.** In practice, $\Delta(\rho)$ can be treated as a tunable calibration hyperparameter. Membership-inference attacks (MIA) can be used as an auditing heuristic to check whether a chosen value leaves any detectable influence from the forget set. Overly small values of $\Delta(\rho)$ lead to detectable leakage, while conservative values suppress the signal. Thus, MIA provides a practical sanity check for selecting a safe $\Delta(\rho)$ for deployment.

## G    DEFINITION OF THE MIA METRIC USED IN TABLE 1

In Table 1 we report the MIA metric exactly as defined by Jia et al. (2023) in Appendix C.3.

**Definition.**    Following their protocol, an MIA model is first trained on (a) a balanced subset of the retained dataset and (b) a separate test dataset (disjoint from the forget set). The trained MIA predictor is then evaluated on the forget set $D_f$ of the unlearned model $\theta_u$.

Let TN be the number of forgotten samples that the MIA predictor classifies as *non-members*. The reported metric is

$$\text{MIA-Efficacy} \;=\; \frac{\text{TN}}{|D_f|}.$$

**Interpretation.**    MIA-Efficacy measures the fraction of forgotten samples that the attacker fails to recognize as training points. Thus, in this convention:

- higher values indicate better unlearning quality;
- MIA-Efficacy = 100% means that all forgotten samples are predicted as non-members.

## H    THE USE OF LARGE LANGUAGE MODELS

We used a large language model (LLM) solely to polish the writing and improve readability. The research ideas, methodology, experiments, and analysis were entirely developed by the authors.

