# OpenReview forum: "Sequential Subspace Noise Injection Prevents Accuracy Collapse in Certified Unlearning"
_ICLR.cc/2026/Conference — Submitted to ICLR 2026_

### Official Review · Reviewer_URSf · 2025-10-27

**Soundness:** 4
**Presentation:** 4
**Contribution:** 4
**Rating:** 6
**Confidence:** 2

**Summary:**

The paper introduces Sequential Subspace Noise Injection to preserve formal certified unlearning guarantees while limiting utility loss. To do this SSNI partitions the model parameter space into orthogonal subspaces and applies noisy fine-tuning sequentially across these blocks. This reduces per-step distortion compared to injecting noise globally. The authors extend the certification framework analysis of noisy fine-tuning to the block-wise setting. And in experiments, SSNI reduces post-unlearning accuracy drop compared to certified baselines for both random and class-wise deletions on MNIST and CIFAR-10 while being robust against membership inference attacks.

**Strengths:**

* SSNI partitions the parameter space into orthogonal subspaces and injects noise sequentially. This reduces per-step distortion and avoids the accuracy collapse from inserting noise all at once to all parameters.

* Empirical results show that SSNI maintains significantly higher post-unlearning accuracy compared to baseline certified methods

* The paper extends prior theory to the block-wise setting and retains the same privacy budget as standard noisy fine-tuning.

**Weaknesses:**

* The method may be memory intensive, but the issue can be reduced by 1) splitting the orthogonal basis layer-wise and 2) using a permutation matrix instead (still layer wise).

**Questions:**

* In practice models with many layers (such as LLMs) may require smaller blocks to reduce memory for the orthogonal subspace decomposition. How does the unlearning - utility tradeoff behave as $k$ gets very large or the number of layers increases dramatically compared to current tested models?

* Are there characteristics of data where SSNI outperforms / underperforms in unlearning compared to the prior unlearning algorithms?

---

> ### Author Response · Authors · 2025-11-24
>
> **W1: The method may be memory intensive, but the issue can be reduced by 1) splitting the orthogonal basis layer-wise and 2) using a permutation matrix instead (still layer wise).**
>
> We agree that storing a full $d \times d$ orthogonal matrix would be memory intensive. In practice, however, we never construct a global basis. As discussed in Sec. 4.3, we operate strictly layer-wise: for a layer with $m \times n$ parameters we either (i) generate an $m \times m$ orthogonal matrix once and cache only its block indices, or (ii) use a permutation matrix representation, which reduces storage to $O(m)$ integers, or (iii) use identity matrices for different layers.
>
> Thus the additional memory footprint is comparable to storing one extra copy of the layer’s weights, and in the permutation case it is typically only a tiny fraction of the layer's storage.
>
> **Q1: In practice models with many layers (such as LLMs) may require smaller blocks to reduce memory for the orthogonal subspace decomposition. How does the unlearning - utility tradeoff behave as $k$ gets very large or the number of layers increases dramatically compared to current tested models?**
>
> We thank the reviewer for pointing this out.
>
> 1. For very deep architectures we do not intend to increase $k$ uniformly across all layers. A natural regime is to use a moderate per-layer value (e.g., $k$=4-8), or to align blocks with existing architectural units such as attention heads or MLP projections. In other words, $k$ *can* be increased if desired, but it does not need to scale linearly with depth.
>
>     To clarify the structure: although each layer is partitioned into $k$ local blocks, the $i$-th blocks across all layers belong to the same *global* block.  Thus the total number of blocks is $k$ (not $k \times \text{number of layers}$), and the unlearning cost scales with $k$ rather than with model depth.
>
> 2. The additional memory introduced by our decomposition depends only on the *layer size*, not on $k$ (as we described in our previous answer). The main effect of a very large $k$ is that the number of unlearning iterations $kT$ increases. Although our experiments show that utility remains stable up to $k=10$, extremely large values would increase runtime. We do not observe meaningful improvements beyond $k\approx 10$ in our current experiments.
>
> **Q2: Are there characteristics of data where SSNI outperforms / underperforms in unlearning compared to the prior unlearning algorithms?**
>
> Across our experiments, SSNI behaves predictably and remains close to full retraining in scenarios where the forget operation induces only a moderate, well-structured shift in the data distribution (e.g., removing a small subset of samples). In such cases, the retained-only retrain stays near the full-retrain trajectory, and SSNI follows it closely.
>
> Within the scope of our current evaluations, we did not observe dataset-specific failure modes; the differences between unlearning methods mainly reflect the expected trade-offs between efficiency and accuracy. A more fine-grained characterization of the regimes in which different unlearning algorithms excel is an interesting direction for future work.

---

> ### Comment · Area_Chair_iy5X · 2025-11-28
> **A gentle reminder to participate in the author–reviewer discussion**
>
> Thank you once again for your service to ICLR 2026. Now that the authors have submitted their rebuttal, could you please engage in the interactive discussion with them? Your participation would be very helpful to the authors, and they would greatly appreciate it. Please also read the authors’ response together with the other reviews and consider whether the rebuttal or any additional comments influence your assessment of the paper.
>
> Thank you again for your efforts.
>
> Best wishes,
>
> Your AC

---

### Official Review · Reviewer_stsM · 2025-10-29

**Soundness:** 3
**Presentation:** 4
**Contribution:** 3
**Rating:** 4
**Confidence:** 4

**Summary:**

This paper aims to address the utility loss of certified unlearning methods based on differential privacy noisy fine tuning. The paper proposes a sequential subspace noise injection scheme that performs block-wise noisy fine-tuning. In particular, it partitions model parameters into orthogonal blocks and, at each stage, unlearn by updating a single block with Gaussian noise before going to the next. Standard fine-tuning is performed at the end.

Based on the shifted-Renyi framework of prior work, this paper extends the analysis to a block-aware setting and claim that composing per-block Renyi guarantees preserves teh overall privacy budget.

The paper also presents a per-step noise lower bound for noisy fine-tuning, which they use to calibrate noise. In the proposed algorithm, this calibration replaces the initial-clipping threshold $C_{0}$ with a term $\Delta(\rho)$.

They provide experiments on image classification benchmarks. The results show that the block-wise scheme can mitigate accuracy collapse after unlearning while maintraining resistance to membership inference attacks.

**Strengths:**

This paper is easy to follow and the algorithm is lised clearly.

The proposed sequential subspace noisy fine-tuning approach for certified unlearning is original and theoretically solid. The theoretical guarantee makes this approach very promising and have valuable potentials. The authors provide theoretical results to validate the reliability of the proposed approach. They also provide theoretical analysis to explain utility collapse in over-parameterized models and motivates the subspace schedule.

Empirically, on standard image classification benchmarks, the proposed method mitigates accuracy loss after unlearning while maintaining strong resistance to membership inference attacks.

**Weaknesses:**

My main concern is about the unvalidated $\Delta(\rho)$.


The paper's certificates replace the unconditional model-clipping threshold $C_{0}$ from Koloskova et al. with a high-probability proximity $\Delta(\rho)$ between the full-data model and the retained-only retrain, and then calibrate noise by substituting $C_{0}$ by $\Delta(\rho)/2$.

$\Delta(\rho)$ is the key component of the paper's guarantee and calibration. It replaces the standard clipping thresold $C_{0}$ everywhere in the analysis and in the algorithm itself, so the certified noise and step counts scale directly with $\Delta(\rho)$.

The paper states replacing $C_{0}$ by $\Delta(\rho)$ tightens calibration. But the authors explicitly state that this "obliges us to estimate or bound $\Delta(\rho)$ in advance". However, in the experiments, they treat $\Delta(\rho)$ as a tunable parameter and do not estimate $\rho$, and ask readers to interpret results under this assumption.

As a result, the experimental results do not establish the claimed privacy budget $(\epsilon, \delta)$ in general. The claimed results are conditional on an unvalidated proximity $\Delta(\rho)$.


Without a reasonable and/or conservative way to obtain or bound $\Delta(\rho)$ (with error analysis), the empirical section effectively reduces to demonstrating a block-wise noise schedule. This schedule by itself is a noise distribution chocie rather than a usable certified-unlearning method. That is, without $\Delta(\rho)$, it is not a novel and operational contribution beyond existing noise-allocation scheduling ideas.

Without specifying a way to estimate a reliable/reasonable $\Delta(\rho)$, the practical advantage collapses to noise scheduling rather than an operational certified-unlearning method.

**Questions:**

Please refer to Weaknesses.

---

> ### Author Response · Authors · 2025-11-24
>
> We thank the reviewer for emphasizing the theoretical importance of the quantity $\Delta(\rho)$.
>
> Our guarantee holds provided that the chosen $\Delta(\rho)$ upper bounds the actual proximity with probability at least $1 - \rho$. Our contribution is to show that, under this standard condition, block-wise scheduling preserves the same privacy budget while avoiding accuracy collapse.
>
> To address the concerns regarding  $\Delta(\rho)$, we have added a new Appendix F titled *"Bounding the proximity $\Delta(\rho)$: theoretical and empirical perspectives"*. This section clarifies the relationship between our proximity term and the classical literature on argument (parameter) stability.
>
> In the revised appendix, we summarize several regimes where $\Delta(\rho)$ \emph{can} be bounded using existing theory:
> (i) smooth non-convex, convex, and strongly convex objectives (Hardt et al. 2016),  (ii) non-smooth convex objectives (Bassily et al. 2020), and (iii) specific neural architectures for which argument-stability proofs are known (e.g., shallow networks, Lei et al. 2022).
> In all of these settings, the parameter deviation $\|\theta^{(t)} - \theta'^{(t)}\|$ under a single-sample replacement admits an explicit upper bound, which directly upper-bounds our discrepancy $\Delta(\rho)$.
>
> We also clarify that deriving such bounds for *arbitrary* deep models remains an open theoretical problem, and we intentionally avoid making unjustified assumptions. Instead, when a closed-form bound is unavailable, Appendix F outlines a practical calibration procedure: a one-time sensitivity measurement performed per architecture, combined with MIA-based auditing heuristics to check that the chosen $\Delta(\rho)$ is sufficiently conservative and does not leave detectable influence from the forget set.
>
> We believe this addition resolves the theoretical ambiguity surrounding $\Delta(\rho)$ and situates our certificate within the well-established framework of stability-based analyses.

---

> ### Comment · Area_Chair_iy5X · 2025-11-28
> **A gentle reminder to participate in the author–reviewer discussion**
>
> Thank you once again for your service to ICLR 2026. Now that the authors have submitted their rebuttal, could you please engage in the interactive discussion with them? Your participation would be very helpful to the authors, and they would greatly appreciate it. Please also read the authors’ response together with the other reviews and consider whether the rebuttal or any additional comments influence your assessment of the paper.
>
> Thank you again for your efforts.
>
> Best wishes,
>
> Your AC

---

### Official Review · Reviewer_48Q5 · 2025-10-31

**Soundness:** 4
**Presentation:** 4
**Contribution:** 4
**Rating:** 6
**Confidence:** 4

**Summary:**

In this paper, the authors investigate a block-wise variant of noisy fine-tuning for certified unlearning. They observe that injecting noise into all parameters simultaneously leads to a sharp drop in test accuracy, which only gradually recovers to the retrained model’s performance. To address this, they propose partitioning the parameter space into orthogonal subspaces and injecting noise progressively across these blocks. This strategy preserves the $(\epsilon, \delta)$-certifiability of unlearning while maintaining test accuracy significantly better than standard noisy fine-tuning approaches.

**Strengths:**

1. The paper is very well written, with a clear and thoughtful analytical presentation.

2. The theoretical results are rigorously proved, and the authors carefully address limitations in prior theorem assumptions.

3. The supporting experiments are well designed and effectively validate the theoretical insights presented in the analysis.

**Weaknesses:**

1. The experiments are conducted primarily on smaller models such as ResNet, raising concerns about the scalability of the proposed method. It remains unclear how well the approach would translate to large-scale, billion-parameter models, or how practical it would be to perform layer-wise orthogonal decomposition across complex architectures.

2. While the empirical results show minimal accuracy loss for block-wise noisy fine-tuning (Block-NFT), it is not clearly established whether the total fine-tuning cost $kT \ll T_{\text{retrain}}$ is guaranteed in practice. Since achieving minimal loss may require increasing $k$, the efficiency advantage over full retraining becomes less certain.

3. In Table 1, only class 5 from CIFAR is deleted. A more comprehensive evaluation would involve deleting multiple classes and reporting the average performance, which would provide a clearer picture of the method’s overall effectiveness and robustness.

**Questions:**

1. Can the proposed orthogonal decomposition be applied to transformer-based architectures, particularly within the attention blocks? If so, how computationally expensive would this process be compared to its implementation in convolutional networks?

2. Is the Chebyshev inequality (Eqn 3) applied in any of the theorems? I might have overlooked its usage if it was implicitly incorporated in the analysis.

3. In Table 1, the method Sa1UN shows a retraining accuracy (RA) closest to the full retrain, and although its runtime is somewhat higher than the proposed method, it is still reasonable. However, its test accuracy is also notably high — does the test set include samples from the forget set, potentially inflating the reported accuracy?

4. Regarding the Membership Inference Attack (MIA) metric — how exactly is it computed in this paper? Ideally, shouldn’t the MIA accuracy be close to 50% (indicating that the attacker cannot reliably distinguish between samples from the forget set and the test set), rather than approaching 100%?

---

> ### Author Response · Authors · 2025-11-24
>
> **W1: The experiments are conducted primarily on smaller models such as ResNet, raising concerns about the scalability of the proposed method. It remains unclear how well the approach would translate to large-scale, billion-parameter models, or how practical it would be to perform layer-wise orthogonal decomposition across complex architectures.**
>
> We thank the reviewer for raising this important question.
>
> (1) The proposed decomposition is not specific to convolutional networks. It is an algebraic construction that can be applied to *any* parameter tensor. For each module, the parameters can be flattened into a vector of dimension $d_{\text{layer}}$, and the orthogonal blocks are constructed in $\mathbb{R}^{d_{\text{layer}}}$ exactly as in the convolutional case. This makes the method architecture-agnostic.
>
> When the layer admits a natural matrix shape $W \in \mathbb{R}^{m \times n}$ with $mn = d_{\text{layer}}$, one may instead perform the construction in $\mathbb{R}^m$ (as in our convolutional and linear layers), which can be computationally more convenient.
>
> (2) To illustrate this generality, we have added experiments with a *transformer* model (ViT-Tiny) on CIFAR-10 (Appendix E.6 in the revised version). In these experiments, we use *exactly the same block-partitioning strategy and the same number of blocks* as in our ResNet experiments. The block decomposition is applied directly to the linear projections in the attention mechanism and to the MLP layers, without any modification of the algorithm.
>
> (3) At larger scales, additional architectural structure (such as sparsity patterns or groups of parameters known to remain nearly static) can be used to choose more targeted blocks, which further improves scalability while leaving the core method unchanged.
>
> **W2: While the empirical results show minimal accuracy loss for block-wise noisy fine-tuning (Block-NFT), it is not clearly established whether the total fine-tuning cost $kT \ll T_{retrain}$. is guaranteed in practice. Since achieving minimal loss may require increasing $k$, the efficiency advantage over full retraining becomes less certain.**
>
> In our experiments we use $T=2$ noisy iterations per block, followed by less than 1.5 additional epochs of standard fine-tuning. Even for the largest $k$ used in our study, the cost of the noisy part ($kT$) is small compared to the post-unlearning fine-tuning, and the overall runtime remains far below that of full retraining (tens of epochs).
>
> The total cost of our method scales with $kT$, where $k$ is the number of blocks. Importantly, $k$ does not need to grow linearly with the number of layers or with model size: in practice, blocks are defined at the architectural level (e.g., grouping corresponding subspaces across layers), so several layers may share the same global block. This keeps $k$ modest even for deep architectures.
>
> In practice, this means that the dominant computational cost of our method is the short fine-tuning phase, not the block-wise noisy updates. As long as $k$ is chosen at the architectural level rather than per-layer, the relation $kT \ll T_{\mathrm{retrain}}$ is naturally maintained without requiring $k$ to scale too much with model size.
>
> **W3: In Table 1, only class 5 from CIFAR is deleted. A more comprehensive evaluation would involve deleting multiple classes and reporting the average performance, which would provide a clearer picture of the method’s overall effectiveness and robustness.**
>
> We thank the reviewer for this suggestion and have added results for deleting all CIFAR-10 classes to the Appendix E.5.

---

> > ### Author Response · Authors · 2025-11-24
> >
> > **Q1: Can the proposed orthogonal decomposition be applied to transformer-based architectures, particularly within the attention blocks? If so, how computationally expensive would this process be compared to its implementation in convolutional networks?**
> >
> > Yes - the orthogonal decomposition applies directly to transformer-based architectures, including attention blocks.
> >
> > Each attention projection (queries, keys, values, output) is a linear operator
> >     with weight matrix $W$, and we use exactly the same block-wise construction as
> >     for convolutional and fully connected layers. In the revised version, we
> >     demonstrate this on a transformer model (ViT-Tiny) on CIFAR-10 in
> >     Appendix E.6 ("Experiments on ViT-Tiny"), where we apply the decomposition
> >     to all linear projections in the attention and MLP layers.
> >
> > **Computational cost vs. convolutional networks.**
> >
> > The cost of the decomposition depends only on the \emph{shape of the weight
> >     matrix}, not on whether the layer is convolutional or part of a transformer.
> >     For a Conv2d layer, the kernel is reshaped into an
> >     $(\text{out}) \times (\text{in}\cdot k_h \cdot k_w)$ matrix; for an attention
> >     projection, it is an $(\text{out}) \times (\text{in})$ matrix. In both cases,
> >     the overhead consists of the same two one-time operations:
> >
> >
> > 1) *Generating an orthonormal basis* $A_1,\ldots,A_m$ for the input
> >     dimension of the layer. This depends only on $m$ and is identical in cost for
> >     a convolutional and a transformer layer with the same input dimension.
> >
> >
> > 2) *Decomposing the weight matrix once* by computing
> >
> >     $W = \sum_i A_i B_i, \qquad B_i = A_i^\top W.$
> >
> >     This is a single matrix multiplication $A^\top W$ per layer, with complexity comparable to one additional forward pass through that layer. After this step, the model is reparameterized, and unlearning proceeds without any further decomposition.
> >
> > Both steps are performed only once as an offline preprocessing stage and do not
> >     depend on the forget set. After unlearning, the original architecture is
> >     recovered by reconstructing $W$ from the stored blocks.
> >
> > During the unlearning stage itself, we update only the block parameters $B_i$.
> >     Each noisy update has essentially the same cost as a standard fine-tuning step,
> >     and is often cheaper because it affects only a fraction of the parameters per
> >     step.
> >
> > Thus, in practice, the computational cost of applying our method to transformers is comparable to its cost in convolutional networks.
> >
> > **Q2: Is the Chebyshev inequality (Eqn 3) applied in any of the theorems? I might have overlooked its usage if it was implicitly incorporated in the analysis.**
> >
> > Eq. (3) is used only to define the high-probability discrepancy $\Delta(\rho)$; the subsequent theorems do not apply Chebyshev directly. Once this discrepancy is fixed, we replace the generic upper bound $2C_0$ on the distance between the retrain-from-scratch model and the original fully trained model with our tighter quantity $\Delta(\rho)$ in the divergence bounds.
> >
> > **Q3: In Table 1, the method Sa1UN shows a retraining accuracy (RA) closest to the full retrain, and although its runtime is somewhat higher than the proposed method, it is still reasonable. However, its test accuracy is also notably high — does the test set include samples from the forget set, potentially inflating the reported accuracy?**
> >
> > Thank you for the question - we clarify the setup. In the class-wise CIFAR-10 experiments, the *forget set* consists of all training samples belonging to one class. After deletion, the new training set contains only the remaining 9 classes.
> >
> > The test set, however, is the standard CIFAR-10 test set and therefore still contains all 10 classes. It does not contain the specific forgotten training samples, but it *does* include test images of the forgotten class.
> >
> > Therefore the retrain-from-scratch model reaches only 86.14% accuracy, which is consistent with the results.
> >
> > **Q4: Regarding the Membership Inference Attack (MIA) metric - how exactly is it computed in this paper? Ideally, shouldn’t the MIA accuracy be close to 50% (indicating that the attacker cannot reliably distinguish between samples from the forget set and the test set), rather than approaching 100%?**
> >
> > We understand the source of confusion.  In Table 1 we follow the MIA metric from *Model Sparsity Can Simplify Machine Unlearning* (Jia et al., 2023). It is defined as the percentage of forgotten samples correctly *hidden* from the attacker:
> >
> > $\text{MIA-Efficacy} = \dfrac{TN}{|D_f|}.$
> >
> > Thus higher is better, and 100\% corresponds to the attacker performing at chance level. We clarify this definition in the Appendix G of revised version.

---

> > > ### Author Response · Authors · 2025-11-24
> > >
> > > We would also like to highlight that the revised version incorporates two substantial additions made directly in response to the reviewer’s main concerns.
> > >
> > > (1) To address the scalability question raised in Weakness 1 and Question 1,
> > > we added transformer-based unlearning experiments (ViT-Tiny) in Appendix E.6. These experiments demonstrate that the proposed block-wise decomposition applies naturally to attention layers and remains efficient in non-convolutional architectures.
> > >
> > > (2) To address Weakness 3, we extended the CIFAR-10 study to all ten classes; the full per-class results are provided in Appendix E.5, giving a more comprehensive view of the method’s robustness.
> > >
> > > We hope these additions clarify the generality and practicality of the approach and would be grateful if the reviewer could consider them when revisiting the overall assessment.

---

> ### Comment · Area_Chair_iy5X · 2025-11-28
> **A gentle reminder to participate in the author–reviewer discussion.**
>
> Thank you once again for your service to ICLR 2026. Now that the authors have submitted their rebuttal, could you please engage in the interactive discussion with them? Your participation would be very helpful to the authors, and they would greatly appreciate it. Please also read the authors’ response together with the other reviews and consider whether the rebuttal or any additional comments influence your assessment of the paper.
>
> Thank you again for your efforts.
>
> Best wishes,
>
> Your AC

---

### Official Review · Reviewer_W5qX · 2025-11-01

**Soundness:** 2
**Presentation:** 3
**Contribution:** 3
**Rating:** 4
**Confidence:** 3

**Summary:**

This paper addresses the practical limitations of certified unlearning approaches based on differential privacy. The core insight is partitioning the parameter space into multiple orthogonal subspaces and sequentially applying noise injection and fine-tuning within each subspace. Empirical validation on the datasets demonstrates the effectiveness of the proposed approach.

**Strengths:**

1.	The paper identifies the limitations of current certified unlearning based on differential privacy and validates the root cause through both theoretical analyses.
2.	The idea of decomposing the noise injection process into orthogonal subspaces and applying it sequentially can preserve the theoretical guarantees of certified unlearning while alleviating the accuracy degradation issue.
3.	Experiments on standard image classification benchmarks consistently demonstrate the efficacy of the proposed approach.

**Weaknesses:**

1. The paper assumes that the parameter space can be partitioned into strictly orthogonal subspaces. Please explain how to correctly distinguish orthogonal parameter subspaces, and why exactly K subspaces can always be partitioned?
2. The parameter space of deep neural networks exhibits high non-convexity and strong coupling characteristics. Could you explain how strong assumptions impact the practical scenario?
3. The proposed approach critically depends on knowing the upper bound $\Delta(\rho)$ of the distance between the initial model and the retrained model. In practical unlearning scenarios, this bound is fundamentally unavailable.
4. In Figure 1, why is there a performance gap between the proposed scheme and the baseline (NFT) unlearning initializations? A similar phenomenon can be observed in Figure 3.
5. In the experiment, options such as K=2, 4, and 10 were used. Is the final choice of K better when larger, smaller, or is there a trade-off? Alternatively, what external factors might influence the selection of K?

**Questions:**

I would appreciate the authors’ responses to the five weaknesses outlined above.

---

> ### Author Response · Authors · 2025-11-24
>
> **Q1: The paper assumes that the parameter space can be partitioned into strictly orthogonal subspaces. Please explain how to correctly distinguish orthogonal parameter subspaces, and why exactly K subspaces can always be partitioned?**
>
> We thank the reviewer for raising this point. To clarify, our method *does not assume* that the model already possesses orthogonal parameter subspaces. Instead, the subspaces are *constructed explicitly*, and this can be done for *any* model and *any* chosen number of subspaces $K$.
>
> As described in Sec. 4.3, for each layer with weight matrix $W \in \mathbb{R}^{m \times n}$ we form an orthonormal basis of $\mathbb{R}^m$ (e.g., via QR decomposition or a permutation matrix). Once an orthonormal basis $A_i$ for  $i = 1,\dots,m$ is obtained, we partition these vectors into $K$ disjoint groups (possibly of different sizes and it is allowed to have empty groups). The spans of the groups define $K$ mutually orthogonal subspaces.
>
> This construction is purely algebraic and does not rely on any assumptions about the loss function or parameter coupling. In practice, we implement it layer-wise using random orthogonal or permutation matrices, as discussed in Sec. 4.3.
>
> **Q2: The parameter space of deep neural networks exhibits high non-convexity and strong coupling characteristics. Could you explain how strong assumptions impact the practical scenario?**
>
> We appreciate the reviewer’s question. Our analysis does *not* rely on convexity of the loss, nor on any decoupling of the parameters across blocks. We follow the noisy fine-tuning framework of Koloskova et al. (2025), which applies to arbitrary (potentially highly non-convex) networks.
>
> The orthogonal decomposition is used solely as an algebraic device to redistribute isotropic Gaussian noise across coordinate directions; it does not impose any structural assumptions on the model or the optimization landscape. Thus, strong coupling between parameters does not affect the validity of our guarantee.
>
> **Q3: The proposed approach critically depends on knowing the upper bound $\Delta(\rho)$ of the distance between the initial model and the retrained model. In practical unlearning scenarios, this bound is fundamentally unavailable.**
>
> We thank the reviewer for highlighting this point. In the revised version we have added Appendix F, which discusses theoretical and practical ways to obtain a bound on $\Delta(\rho)$.
>
> Our method does not rely on having a closed-form analytical expression for $\Delta(\rho)$ in arbitrary deep networks. Instead, the framework allows $\Delta(\rho)$ to be obtained in two complementary ways.
>
> (1) *In several standard settings* - including $\beta$-smooth non-convex losses, nonsmooth convex losses, and specific neural architectures -- prior work on argument stability provides explicit upper bounds on the parameter deviation between two SGD trajectories trained on neighboring datasets. In these regimes, $\Delta(\rho)$ can be instantiated directly from existing theory.
>
> (2) *For general deep models*, for which obtaining a closed-form bound is a known open problem, $\Delta(\rho)$ can be treated as a model-level proximity parameter that is fixed for a task. In Appendix F we discuss practical ways to obtain such a conservative value, including sensitivity-based calibration and other stability-inspired estimators. We also use membership-inference attacks (MIA) as an auditing heuristic to check that a chosen $\Delta(\rho)$ does not leave residual influence from the forget set.
>
> Thus, while a universal closed-form formula is not available, the method remains operational: $\Delta(\rho)$ can be obtained from theory when available, or fixed conservatively and checked through auditing when not.
>
> **Q4: In Figure 1, why is there a performance gap between the proposed scheme and the baseline (NFT) unlearning initializations? A similar phenomenon can be observed in Figure 3.**
>
> We thank the reviewer for the question. To avoid confusion, we emphasize that both NFT and Block-NFT start from *exactly the same* fully trained model $\hat{\mathbf{x}}$; there is no difference in initialization.
>
> The observed gap appears \textbf{during} the first unlearning steps, and we understand why this may look like an initialization difference. In both figures, unlearning consists of only two iterations of noisy fine-tuning per block, while all subsequent iterations are regular fine-tuning. Because these noisy updates immediately perturb the parameters in different ways for NFT and Block-NFT, the trajectories diverge almost instantly. Thus, even though the initializations are identical, a noticeable difference is visible after just a few steps.

---

> > ### Author Response · Authors · 2025-11-24
> >
> > **Q5: In the experiment, options such as $K$=2, 4, and 10 were used. Is the final choice of $K$ better when larger, smaller, or is there a trade-off? Alternatively, what external factors might influence the selection of $K$?**
> >
> > There is indeed a trade-off in the choice of $K$. Increasing $K$ reduces the number of perturbed coordinates per step (only $1/K$ of parameters are updated), which leads to smaller per-step distortion and smoother optimization trajectories (see Fig. 3). On the other hand, the number of block updates scales linearly with $K$.
> >
> > Empirically, we find that increasing $K$ beyond roughly 10 yields little additional benefit on MNIST and CIFAR-10. For models of the scale of ResNet-18, a practical choice is to use $K$ = 4 - 10.

---

> > ### Comment · Area_Chair_iy5X · 2025-11-28
> > **A gentle reminder to participate in the author–reviewer discussion**
> >
> > Thank you once again for your service to ICLR 2026. Now that the authors have submitted their rebuttal, could you please engage in the interactive discussion with them? Your participation would be very helpful to the authors, and they would greatly appreciate it. Please also read the authors’ response together with the other reviews and consider whether the rebuttal or any additional comments influence your assessment of the paper.
> >
> > Thank you again for your efforts.
> >
> > Best wishes,
> >
> > Your AC

---

### Author Response · Authors · 2025-11-24
**Comment on the revision**

We highlight the main updates included in the revised version.

**(1) Additional experiments.**

We added two empirical components in response to reviewer feedback:

1. Experiments with a transformer-based architecture (ViT-Tiny) on CIFAR-10
(App. E.6), demonstrating that the block-wise decomposition applies directly to non-convolutional models.

2. Full per-class deletion experiments for all 10 classes of CIFAR-10
(App. E.5), complementing the single-class results reported in the main text. We also expanded the appendix with a clarification of the MIA metric used in
our evaluation (App. G).

**(2) Discussion of $\Delta(\rho)$.**

We added Appendix F, which provides a discussion of the proximity parameter $\Delta(\rho)$. This appendix summarizes how existing argument-stability results can be used  to upper-bound parameter deviation in several standard training regimes, and outlines how $\Delta(\rho)$ can be handled in practice when working with general deep architectures.

---

### Author Response · Authors · 2025-11-28

Dear AC,

We wanted to kindly ask if you could remind the reviewers to respond to our revision in the current discussion period. We are hoping to receive timely feedback, as we have incorporated several updates and believe further input would greatly benefit the quality of our work and the review process. Thank you very much for your assistance and support.

Best regards,

The Authors

---

> ### Comment · Area_Chair_iy5X · 2025-11-28
> **Re: Official Comment by Authors**
>
> Dear Authors,
>
> Thank you for reaching out. We have just sent individual reminders to all reviewers to encourage them to engage in the discussion.
>
> Best wishes,
>
> Your AC

---

### Meta-Review · Area_Chair_HuDv · 2025-12-27

**Summary:**

The main issue of the paper is that the work is hinged on $\Delta(\rho)$ although it is not validated nor bounded (in practice.) Both reviewers W5qX and stsM pointed this out, and I agree with them. Another is the scalability issue of the approach. The authors did not make a convincing argument on it. Hence, I do not recommend accepting this paper.

**Reviewer Concerns:**

* Reviewer W5qX:  The reviewer’s Q3 was valid but the response to Q3 is not fully convincing if the approach will be operational in practice.
For Q5, the authors could have provided comprehensive explorations regarding K.

* Reviewer 48Q5: W1: The reviewer’s concern about scalability toward large-scale models has not been addressed. The additional experiment which was done on ViT-Tiny is still not a large enough model, and the authors’ response (3) does not address it. By considering that the authors mentioned “sparsity patterns” and “group of parameters”, I infer that the approach may work only when the parameters form specific sparsity patterns or contain a limited number of groups. If this is true, the reviewer’s concern is still outstanding, and I agree with the reviewer’s concern. Additionally, this concern is evident in the author’s response to Reviewer stsM, stating that deriving such bounds for arbitrary deep models remains an open theoretical problem.

* Reviewer stsM: The same concern with Q3 of Reviewer W5qX was raised. The concern and the supporting arguments of the reviewer is valid, but the response was not fully convincing. I agree with the reviewer that because of the issue, the contribution of the work boils down to noise scheduling.

**Reviewer Scores:**

* Reviewer W5qX: I think the reviewer would not have changed the score.
* Reviewer 48Q5: I think the reviewer would not have changed the score.
* Reviewer stsM: I think the reviewer would not have changed the score.
* Reviewer URSf: I think the reviewer would not have changed the score.

---

### Decision · Program_Chairs · 2026-01-26

Reject